# TimesNet: Temporal 2D-Variation Modeling for General Time Series Analysis

**Haixu Wu,**[*] **Tengge Hu,**[*] **Yong Liu,**[*] **Hang Zhou, Jianmin Wang, Mingsheng Long**[✉]
School of Software, BNRist, Tsinghua University, Beijing 100084, China
`{whx20,liuyong21,htg21,h-zhou18}@mails.tsinghua.edu.cn`
`{jimwang,mingsheng}@tsinghua.edu.cn`

## Abstract

Time series analysis is of immense importance in extensive applications, such as weather forecasting, anomaly detection, and action recognition. This paper focuses on temporal variation modeling, which is the common key problem of extensive analysis tasks. Previous methods attempt to accomplish this directly from the 1D time series, which is extremely challenging due to the intricate temporal patterns. Based on the observation of multi-periodicity in time series, we ravel out the complex temporal variations into the multiple *intraperiod-* and *interperiod-variations*. To tackle the limitations of 1D time series in representation capability, we extend the analysis of temporal variations into the 2D space by transforming the 1D time series into a set of 2D tensors based on multiple periods. This transformation can embed the intraperiod- and interperiod-variations into the columns and rows of the 2D tensors respectively, making the 2D-variations to be easily modeled by 2D kernels. Technically, we propose the *TimesNet* with *TimesBlock* as a task-general backbone for time series analysis. TimesBlock can discover the multi-periodicity adaptively and extract the complex temporal variations from transformed 2D tensors by a parameter-efficient inception block. Our proposed TimesNet achieves consistent state-of-the-art in five mainstream time series analysis tasks, including short- and long-term forecasting, imputation, classification, and anomaly detection. Code is available at this repository: https://github.com/thuml/TimesNet.

## 1 Introduction

Time series analysis is widely used in extensive real-world applications, such as the forecasting of meteorological factors for weather prediction (Wu et al., 2021), imputation of missing data for data mining (Friedman, 1962), anomaly detection of monitoring data for industrial maintenance (Xu et al., 2021) and classification of trajectories for action recognition (Franceschi et al., 2019). Because of its immense practical value, time series analysis has received great interest (Lim & Zohren, 2021).

Different from other types of sequential data, such as language or video, time series is recorded continuously and each time point only saves some scalars. Since one single time point usually cannot provide sufficient semantic information for analysis, many works focus on the temporal variation, which is more informative and can reflect the inherent properties of time series, such as the continuity, periodicity, trend and etc. However, the variations of real-world time series always involve intricate temporal patterns, where multiple variations (e.g. rising, falling, fluctuation and etc.) mix and overlap with each other, making the temporal variation modeling extremely challenging.

Especially in the deep learning communities, benefiting from the powerful non-linear modeling capacity of deep models, many works have been proposed to capture the complex temporal variations in real-world time series. One category of methods adopts recurrent neural networks (RNN) to model the successive time points based on the Markov assumption (Hochreiter & Schmidhuber, 1997; Lai et al., 2018; Shen et al., 2020). However, these methods usually fail in capturing the long-term dependencies and their efficiency suffers from the sequential computation paradigm. Another category of methods utilizes the convolutional neural network along the temporal dimension (TCN)

---

[*]Equal Contribution

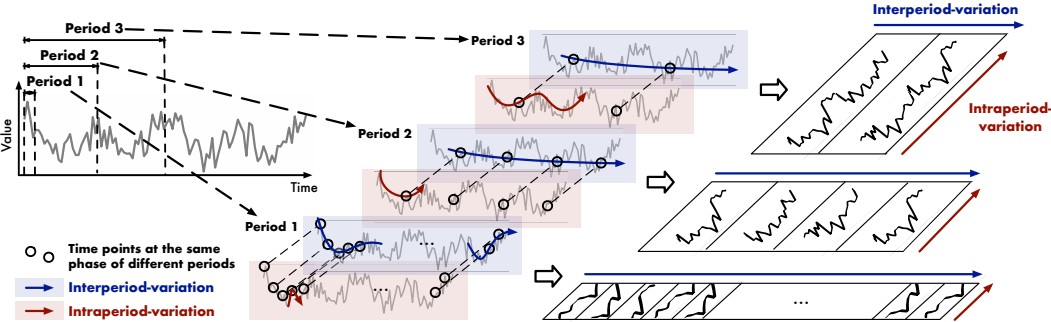

Figure 1: Multi-periodicity and temporal 2D-variation of time series. Each period involves the intraperiod-variation and interperiod-variation. We transform the original 1D time series into a set of 2D tensors based on multiple periods, which can unify the intraperiod- and interperiod-variations.

to extract the variation information (Franceschi et al., 2019; He & Zhao, 2019). Also, because of the locality property of the one-dimension convolution kernels, they can only model the variations among adjacent time points, thereby still failing in long-term dependencies. Recently, Transformers with attention mechanism have been widely used in sequential modeling (Brown et al., 2020; Dosovitskiy et al., 2021; Liu et al., 2021b). In time series analysis, many Transformer-based models adopt the attention mechanism or its variants to capture the pair-wise temporal dependencies among time points (Li et al., 2019; Kitaev et al., 2020; Zhou et al., 2021; 2022). But it is hard for attention mechanism to find out reliable dependencies directly from scattered time points, since the temporal dependencies can be obscured deeply in intricate temporal patterns (Wu et al., 2021).

In this paper, to tackle the intricate temporal variations, we analyze the time series from a new dimension of multi-periodicity. Firstly, we observe that real-world time series usually present multi-periodicity, such as daily and yearly variations for weather observations, weekly and quarterly variations for electricity consumption. These multiple periods overlap and interact with each other, making the variation modeling intractable. Secondly, for each period, we find out that the variation of each time point is not only affected by the temporal pattern of its adjacent area but also highly related to the variations of its adjacent periods. For clearness, we name these two types of temporal variations as *intraperiod-variation* and *interperiod-variation* respectively. The former indicates short-term temporal patterns within a period. The latter can reflect long-term trends of consecutive different periods. Note that for the time series without clear periodicity, the variations will be dominated by the intraperiod-variation and is equivalent to the ones with infinite period length.

Since different periods will lead to different intraperiod- and interperiod-variations, the multi-periodicity can naturally derive a modular architecture for temporal variation modeling, where we can capture the variations derived by a certain period in one module. Besides, this design makes the intricate temporal patterns disentangled, benefiting the temporal variation modeling. However, it is notable that the 1D time series is hard to explicitly present two different types of variations simultaneously. To tackle this obstacle, we extend the analysis of temporal variations into the 2D space. Concretely, as shown in Figure 1, we can reshape the 1D time series into a 2D tensor, where each column contains the time points within a period and each row involves the time points at the same phase among different periods. Thus, by transforming 1D time series into a set of 2D tensors, we can break the bottleneck of representation capability in the original 1D space and successfully unify the intraperiod- and interperiod-variations in 2D space, obtaining the *temporal 2D-variations*.

Technically, based on above motivations, we go beyond previous backbones and propose the *TimesNet* as a new task-general model for time series analysis. Empowering by *TimesBlock*, TimesNet can discover the multi-periodicity of time series and capture the corresponding temporal variations in a modular architecture. Concretely, TimesBlock can adaptively transform the 1D time series into a set of 2D tensors based on learned periods and further capture intraperiod- and interperiod-variations in the 2D space by a parameter-efficient inception block. Experimentally, TimesNet achieves the consistent state-of-the-art in five mainstream analysis tasks, including short- and long-term forecasting, imputation, classification and anomaly detection. Our contributions are summarized in three folds:

- Motivated by multi-periodicity and complex interactions within and between periods, we find out a modular way for temporal variation modeling. By transforming the 1D time series into 2D space, we can present the intraperiod- and interperiod-variations simultaneously.

- We propose the TimesNet with TimesBlock to discover multiple periods and capture temporal 2D-variations from the transformed 2D tensors by a parameter-efficient inception block.

- As a task-general foundation model, TimesNet achieves the consistent state-of-the-art in five mainstream time series analysis tasks. Detailed and insightful visualizations are included.

## 2 RELATED WORK

As a key problem of time series analysis, temporal variation modeling has been well explored.

Many classical methods assume that the temporal variations follow the pre-defined patterns, such as ARIMA (Anderson & Kendall, 1976), Holt-Winter (Hyndman & Athanasopoulos, 2018) and Prophet (Taylor & Letham, 2018). However, the variations of real-world time series are usually too complex to be covered by pre-defined patterns, limiting the practical applicability of these classical methods.

In recent years, many deep models have been proposed for temporal modeling, such as MLP, TCN, RNN-based models (Hochreiter & Schmidhuber, 1997; Lai et al., 2018; Franceschi et al., 2019). Technically, MLP-based methods (Oreshkin et al., 2019; Challu et al., 2022; Zeng et al., 2023; Zhang et al., 2022) adopt the MLP along the temporal dimension and encode the temporal dependencies into the fixed parameter of MLP layers. The TCN-based (2019) methods capture the temporal variations by convolutional kernels that slide along the temporal dimension. The RNN-based methods (Hochreiter & Schmidhuber, 1997; Lai et al., 2018; Gu et al., 2022) utilize the recurrent structure and capture temporal variations implicitly by state transitions among time steps. Note that none of these methods consider the temporal 2D-variations derived by periodicity, which is proposed in this paper.

Besides, Transformers have shown great performance in time series forecasting (Zhou et al., 2021; Liu et al., 2021a; Wu et al., 2021; Zhou et al., 2022). With attention mechanism, they can discover the temporal dependencies among time points. Especially, Wu et al. present the Autoformer with Auto-Correlation mechanism to capture the series-wise temporal dependencies based on the learned periods. In addition, to tackle the intricate temporal patterns, Autoformer also presents a deep decomposition architecture to obtain the seasonal and trend parts of input series. Afterward, FEDformer (Zhou et al., 2022) employs the mixture-of-expert design to enhance the seasonal-trend decomposition and presents a sparse attention within the frequency domain. Unlike previous methods, we ravel out the intricate temporal patterns by exploring the multi-periodicity of time series and capture the temporal 2D-variations in 2D space by well-acknowledged computer vision backbones for the first time.

It is also notable that, different from previous methods, we no longer limit to a specific analysis task and attempt to propose a task-general foundation model for time series analysis.

## 3 TIMESNET

As aforementioned, based on the multi-periodicity of time series, we propose the *TimesNet* with a modular architecture to capture the temporal patterns derived from different periods. For each period, to capture the corresponding intraperiod- and interperiod-variations, we design a *TimesBlock* within the TimesNet, which can transform the 1D time series into 2D space and simultaneously model the two types of variations by a parameter-efficient inception block.

### 3.1 TRANSFORM 1D-VARIATIONS INTO 2D-VARIATIONS

As shown in Figure 1, each time point involves two types of temporal variations with its adjacent area and with the same phase among different periods simultaneously, namely *intraperiod-* and *interperiod-variations*. However, this original 1D structure of time series can only present the variations among adjacent time points. To tackle this limitation, we explore the two-dimension structure for temporal variations, which can explicitly present variations within and between periods, thereby with more advantages in representation capability and benefiting the subsequent representation learning.

Concretely, for the length-$T$ time series with $C$ recorded variates, the original 1D organization is $\mathbf{X}_{\mathrm{1D}} \in \mathbb{R}^{T \times C}$. To represent the interperiod-variation, we need to discover periods first. Technically, we analyze the time series in the frequency domain by Fast Fourier Transform (FFT) as follows:

$$\mathbf{A} = \mathrm{Avg}\left(\mathrm{Amp}\left(\mathrm{FFT}(\mathbf{X}_{\mathrm{1D}})\right)\right), \{f_1, \cdots, f_k\} = \underset{f_* \in \{1, \cdots, [\frac{T}{2}]\}}{\arg \mathrm{Topk}} (\mathbf{A}), \ p_i = \left\lceil \frac{T}{f_i} \right\rceil, i \in \{1, \cdots, k\}.$$

$$(1)$$

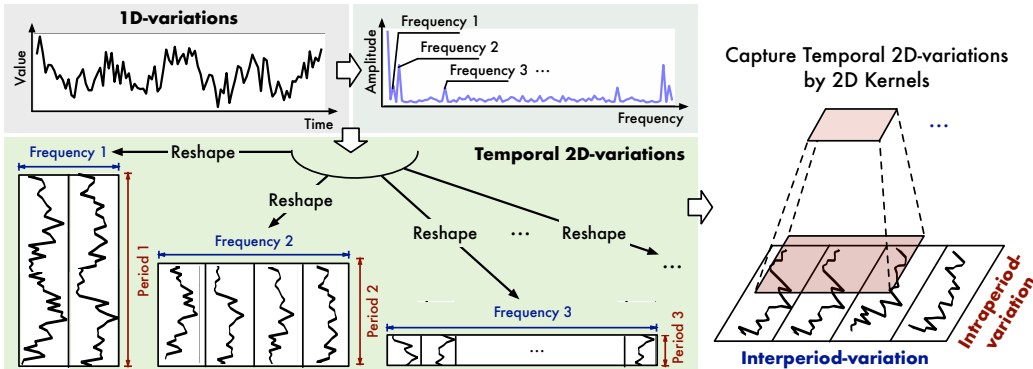

Figure 2: A univariate example to illustrate 2D structure in time series. By discovering the periodicity, we can transform the original 1D time series into structured 2D tensors, which can be processed by 2D kernels conveniently. By conducting the same reshape operation to all variates of time series, we can extend the above process to multivariate time series.

Here, $\text{FFT}(\cdot)$ and $\text{Amp}(\cdot)$ denote the FFT and the calculation of amplitude values. $\mathbf{A} \in \mathbb{R}^T$ represents the calculated amplitude of each frequency, which is averaged from $C$ dimensions by $\text{Avg}(\cdot)$. Note that the $j$-th value $\mathbf{A}_j$ represents the intensity of the frequency-$j$ periodic basis function, corresponding to the period length $\lceil \frac{T}{j} \rceil$. Considering the sparsity of frequency domain and to avoid the noises brought by meaningless high frequencies (Chatfield, 1981; Zhou et al., 2022), we only select the top-$k$ amplitude values and obtain the most significant frequencies $\{f_1, \cdots, f_k\}$ with the unnormalized amplitudes $\{\mathbf{A}_{f_1}, \cdots, \mathbf{A}_{f_k}\}$, where $k$ is the hyper-parameter. These selected frequencies also correspond to $k$ period lengths $\{p_1, \cdots, p_k\}$. Due to the conjugacy of frequency domain, we only consider the frequencies within $\{1, \cdots, [\frac{T}{2}]\}$. We summarize Equation 1 as follows:

$$\mathbf{A}, \{f_1, \cdots, f_k\}, \{p_1, \cdots, p_k\} = \text{Period}(\mathbf{X}_{\text{1D}}). \tag{2}$$

Based on the selected frequencies $\{f_1, \cdots, f_k\}$ and corresponding period lengths $\{p_1, \cdots, p_k\}$, we can reshape the 1D time series $\mathbf{X}_{\text{1D}} \in \mathbb{R}^{T \times C}$ into multiple 2D tensors by the following equations:

$$\mathbf{X}_{\text{2D}}^i = \text{Reshape}_{p_i, f_i}\left(\text{Padding}(\mathbf{X}_{\text{1D}})\right), \ i \in \{1, \cdots, k\}, \tag{3}$$

where $\text{Padding}(\cdot)$ is to extend the time series by zeros along temporal dimension to make it compatible for $\text{Reshape}_{p_i, f_i}(\cdot)$, where $p_i$ and $f_i$ represent the number of rows and columns of the transformed 2D tensors respectively. Note that $\mathbf{X}_{\text{2D}}^i \in \mathbb{R}^{p_i \times f_i \times C}$ denotes the $i$-th reshaped time series based on frequency-$f_i$, whose columns and rows represent the intraperiod-variation and interperiod-variation under the corresponding period length $p_i$ respectively. Eventually, as shown in Figure 2, based on the selected frequencies and estimated periods, we obtain a set of 2D tensors $\{\mathbf{X}_{\text{2D}}^1, \cdots, \mathbf{X}_{\text{2D}}^k\}$, which indicates $k$ different temporal 2D-variations derived by different periods.

It is also notable that, this transformation brings two types of localities to the transformed 2D tensors, that is localities among adjacent time points (columns, intraperiod-variation) and adjacent periods (rows, interperiod-variation). Thus, the temporal 2D-variations can be easily processed by 2D kernels.

## 3.2 TIMESBLOCK

As shown in Figure 3, we organize the TimesBlock in a residual way (He et al., 2016). Concretely, for the length-$T$ 1D input time series $\mathbf{X}_{\text{1D}} \in \mathbb{R}^{T \times C}$, we project the raw inputs into the deep features $\mathbf{X}_{\text{1D}}^0 \in \mathbb{R}^{T \times d_{\text{model}}}$ by the embedding layer $\mathbf{X}_{\text{1D}}^0 = \text{Embed}(\mathbf{X}_{\text{1D}})$ at the very beginning. For the $l$-th layer of TimesNet, the input is $\mathbf{X}_{\text{1D}}^{l-1} \in \mathbb{R}^{T \times d_{\text{model}}}$ and the process can be formalized as:

$$\mathbf{X}_{\text{1D}}^l = \text{TimesBlock}\left(\mathbf{X}_{\text{1D}}^{l-1}\right) + \mathbf{X}_{\text{1D}}^{l-1}. \tag{4}$$

As shown in Figure 3, for the $l$-th TimesBlock, the whole process involves two successive parts: capturing temporal 2D-variations and adaptively aggregating representations from different periods.

**Capturing temporal 2D-variations** Similar to Equation 1, we can estimate period lengths for deep features $\mathbf{X}_{\text{1D}}^{l-1}$ by $\text{Period}(\cdot)$. Based on estimated period lengths, we can transform the 1D time series

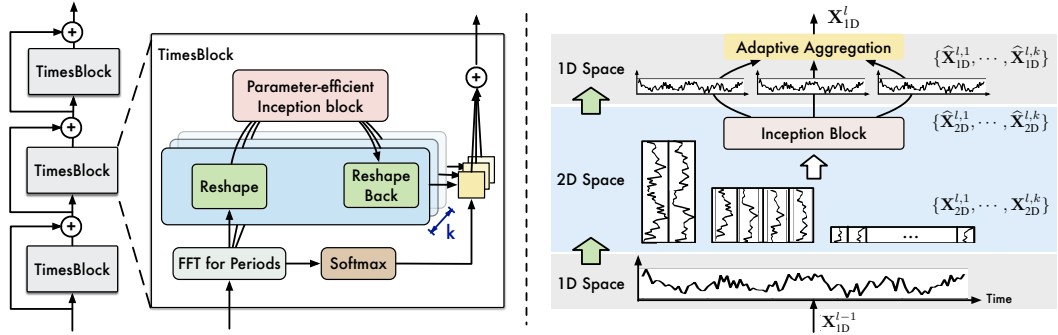

Figure 3: Overall architecture of TimesNet. TimesNet is stacked by TimesBlocks in a residual way. TimesBlocks can capture various temporal 2D-variations from $k$ different reshaped tensors by a parameter-efficient inception block in 2D space and fuse them based on normalized amplitude values.

into 2D space and obtain a set of 2D tensors, from which we can obtain informative representations by parameter-efficient inception block conveniently. The process is formalized as follows:

$$
\begin{aligned}
\mathbf{A}^{l-1}, \{f_1, \cdots, f_k\}, \{p_1, \cdots, p_k\} &= \text{Period}\left(\mathbf{X}_{1D}^{l-1}\right), \\
\mathbf{X}_{2D}^{l,i} &= \text{Reshape}_{p_i, f_i}\left(\text{Padding}(\mathbf{X}_{1D}^{l-1})\right), \; i \in \{1, \cdots, k\} \\
\widehat{\mathbf{X}}_{2D}^{l,i} &= \text{Inception}\left(\mathbf{X}_{2D}^{l,i}\right), \; i \in \{1, \cdots, k\} \\
\widehat{\mathbf{X}}_{1D}^{l,i} &= \text{Trunc}\left(\text{Reshape}_{1,(p_i \times f_i)}\left(\widehat{\mathbf{X}}_{2D}^{l,i}\right)\right), \; i \in \{1, \cdots, k\},
\end{aligned}
\tag{5}
$$

where $\mathbf{X}_{2D}^{l,i} \in \mathbb{R}^{p_i \times f_i \times d_{\text{model}}}$ is the $i$-th transformed 2D tensor. After the transformation, we process the 2D tensor by a parameter-efficient inception block (Szegedy et al., 2015) as $\text{Inception}(\cdot)$, which involves multi-scale 2D kernels and is one of the most well-acknowledged vision backbones. Then we transform the learned 2D representations $\widehat{\mathbf{X}}_{2D}^{l,i}$ back to 1D space $\widehat{\mathbf{X}}_{1D}^{l,i} \in \mathbb{R}^{T \times d_{\text{model}}}$ for aggregation, where we employ $\text{Trunc}(\cdot)$ to truncate the padded series with length $(p_i \times f_i)$ into original length $T$.

Note that benefiting from the transformation of 1D time series, the 2D kernels in the inception block can aggregate the multi-scale intraperiod-variation (columns) and interperiod-variation (rows) simultaneously, covering both adjacent time points and adjacent periods. Besides, we adopt a shared inception block for different reshaped 2D tensors $\{\mathbf{X}_{2D}^{l,1}, \cdots, \mathbf{X}_{2D}^{l,k}\}$ to improve parameter efficiency, which can make the model size invariant to the selection of hyper-parameter $k$.

**Adaptive aggregation** Finally, we need to fuse $k$ different 1D-representations $\{\widehat{\mathbf{X}}_{1D}^{l,1}, \cdots, \widehat{\mathbf{X}}_{1D}^{l,k}\}$ for the next layer. Inspired by Auto-Correlation (Wu et al., 2021), the amplitudes $\mathbf{A}$ can reflect the relative importance of selected frequencies and periods, thereby corresponding to the importance of each transformed 2D tensor. Thus, we aggregate the 1D-representations based on the amplitudes:

$$
\begin{aligned}
\widehat{\mathbf{A}}_{f_1}^{l-1}, \cdots, \widehat{\mathbf{A}}_{f_k}^{l-1} &= \text{Softmax}\left(\mathbf{A}_{f_1}^{l-1}, \cdots, \mathbf{A}_{f_k}^{l-1}\right) \\
\mathbf{X}_{1D}^{l} &= \sum_{i=1}^{k} \widehat{\mathbf{A}}_{f_i}^{l-1} \times \widehat{\mathbf{X}}_{1D}^{l,i}.
\end{aligned}
\tag{6}
$$

Since the variations within and between periods are already involved in multiple highly-structured 2D tensors, TimesBlock can fully capture multi-scale temporal 2D-variations simultaneously. Thus, TimesNet can achieve a more effective representation learning than directly from 1D time series.

**Generality in 2D vision backbones** Benefiting from the transformation of 1D time series into temporal 2D-variations, we can choose various computer vision backbones to replace the inception block for representation learning, such as the widely-used ResNet (He et al., 2016) and ResNeXt (Xie et al., 2017), advanced ConvNeXt (Liu et al., 2022b) and attention-based models (Liu et al., 2021b). Thus, our temporal 2D-variation design also bridges the 1D time series to the booming 2D vision backbones, making the time series analysis take advantage of the development of computer vision community. In general, more powerful 2D backbones for representation learning will bring better performance. Considering both performance and efficiency (Figure 4 right), we conduct the main experiments based on the parameter-efficient inception block as shown in Equation 5.

# 4 EXPERIMENTS

To verify the generality of TimesNet, we extensively experiment on five mainstream analysis tasks, including short- and long-term forecasting, imputation, classification and anomaly detection.

**Implementation** Table 1 is a summary of benchmarks. More details about the dataset, experiment implementation and model configuration can be found in Appendix A.

Table 1: Summary of experiment benchmarks.

| Tasks | Benchmarks | Metrics | Series Length |
|---|---|---|---|
| Forecasting | **Long-term**: ETT (4 subsets), Electricity, Traffic, Weather, Exchange, ILI | MSE, MAE | 96∼720 (ILI: 24∼60) |
| | **Short-term**: M4 (6 subsets) | SMAPE, MASE, OWA | 6∼48 |
| Imputation | ETT (4 subsets), Electricity, Weather | MSE, MAE | 96 |
| Classification | UEA (10 subsets) | Accuracy | 29∼1751 |
| Anomaly Detection | SMD, MSL, SMAP, SWaT, PSM | Precision, Recall, F1-Socre | 100 |

**Baselines** Since we attempt to propose a foundation model for time series analysis, we extensively compare the well-acknowledged and advanced models in all five tasks, including the RNN-based models: LSTM (1997), LSTNet (2018) and LSSL (2022); CNN-based Model: TCN (2019); MLP-based models: LightTS (2022) and DLinear (2023); Transformer-based models: Reformer (2020), Informer (2021), Pyraformer (2021a), Autoformer (2021), FEDformer (2022), Non-stationary Transformer (2022a) and ETSformer (2022). Besides, we also compare the state-of-the-art models for each specific task, such as N-HiTS (2022) and N-BEATS (2019) for short-term forecasting, Anomaly Transformer (2021) for anomaly detection, Rocket (2020) and Flowformer (2022) for classification and etc. Overall, more than 15 baselines are included for a comprehensive comparison.

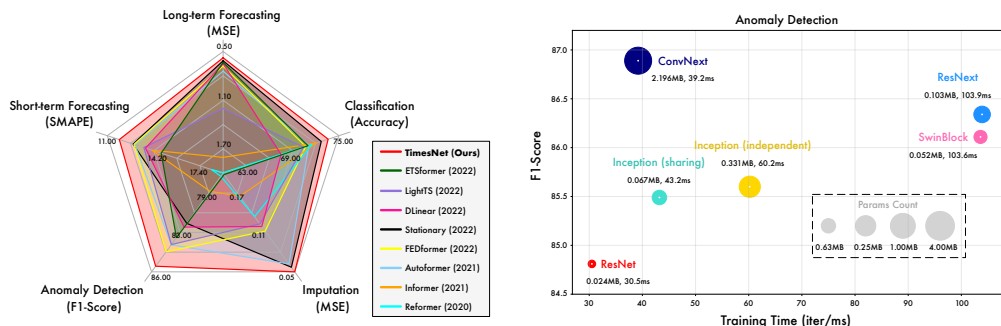

Figure 4: Model performance comparison (left) and generality in different vision backbones (right).

## 4.1 MAIN RESULTS

As a foundation model, TimesNet achieves consistent state-of-the-art performance on five mainstream analysis tasks compared with other customized models (Figure 4 left). The full efficiency comparison is provided in Table 11 of Appendix. Besides, by replacing the inception block with more powerful vision backbones, we can further promote the performance of TimesNet (Figure 4 right), confirming that our design can make time series analysis take advantage of booming vision backbones.

## 4.2 SHORT- AND LONG-TERM FORECASTING

**Setups** Time series forecasting is essential in weather forecasting, traffic and energy consumption planning. To fully evaluate the model performance in forecasting, we adopt two types of benchmarks, including long-term and short-term forecasting. Especially for the long-term setting, we follow the benchmarks used in Autoformer (2021), including ETT (Zhou et al., 2021), Electricity (UCI), Traffic (PeMS), Weather (Wetterstation), Exchange (Lai et al., 2018) and ILI (CDC), covering five real-world applications. For the short-term dataset, we adopt the M4 (Spyros Makridakis, 2018), which contains the yearly, quarterly and monthly collected univariate marketing data. Note that each dataset in the long-term setting only contains one continuous time series, where we obtain samples by sliding window, while M4 involves 100,000 different time series collected in different frequencies.

Table 2: Long-term forecasting task. The past sequence length is set as 36 for ILI and 96 for the others. All the results are averaged from 4 different prediction lengths, that is {24, 36, 48, 60} for ILI and {96, 192, 336, 720} for the others. See Table 13 in Appendix for the full results.

| Models | TimesNet (Ours) | | ETSformer (2022) | | LightTS (2022) | | DLinear (2023) | | FEDformer (2022) | | Stationary (2022a) | | Autoformer (2021) | | Pyraformer (2021a) | | Informer (2021) | | LogTrans (2019) | | Reformer (2020) | |
|---|---|---|---|---|---|---|---|---|---|---|---|---|---|---|---|---|---|---|---|---|---|---|
| Metric | MSE | MAE | MSE | MAE | MSE | MAE | MSE | MAE | MSE | MAE | MSE | MAE | MSE | MAE | MSE | MAE | MSE | MAE | MSE | MAE | MSE | MAE |
| ETTm1 | **0.400** | **0.406** | 0.429 | 0.425 | 0.435 | 0.437 | 0.403 | 0.407 | 0.448 | 0.452 | 0.481 | 0.456 | 0.588 | 0.517 | 0.691 | 0.607 | 0.961 | 0.734 | 0.929 | 0.725 | 0.799 | 0.671 |
| ETTm2 | **0.291** | **0.333** | 0.293 | 0.342 | 0.409 | 0.436 | 0.350 | 0.401 | 0.305 | 0.349 | 0.306 | 0.347 | 0.327 | 0.371 | 1.498 | 0.869 | 1.410 | 0.810 | 1.535 | 0.900 | 1.479 | 0.915 |
| ETTh1 | 0.458 | **0.450** | 0.542 | 0.510 | 0.491 | 0.479 | 0.456 | 0.452 | **0.440** | 0.460 | 0.570 | 0.537 | 0.496 | 0.487 | 0.827 | 0.703 | 1.040 | 0.795 | 1.072 | 0.837 | 1.029 | 0.805 |
| ETTh2 | **0.414** | **0.427** | 0.439 | 0.452 | 0.602 | 0.543 | 0.559 | 0.515 | 0.437 | 0.449 | 0.526 | 0.516 | 0.450 | 0.459 | 0.826 | 0.703 | 4.431 | 1.729 | 2.686 | 1.494 | 6.736 | 2.191 |
| Electricity | **0.192** | **0.295** | 0.208 | 0.323 | 0.229 | 0.329 | 0.212 | 0.300 | 0.214 | 0.327 | 0.193 | 0.296 | 0.227 | 0.338 | 0.379 | 0.445 | 0.311 | 0.397 | 0.272 | 0.370 | 0.338 | 0.422 |
| Traffic | 0.620 | **0.336** | 0.621 | 0.396 | 0.622 | 0.392 | 0.625 | 0.383 | **0.610** | 0.376 | 0.624 | 0.340 | 0.628 | 0.379 | 0.878 | 0.469 | 0.764 | 0.416 | 0.705 | 0.395 | 0.741 | 0.422 |
| Weather | **0.259** | **0.287** | 0.271 | 0.334 | 0.261 | 0.312 | 0.265 | 0.317 | 0.309 | 0.360 | 0.288 | 0.314 | 0.338 | 0.382 | 0.946 | 0.717 | 0.634 | 0.548 | 0.696 | 0.602 | 0.803 | 0.656 |
| Exchange | 0.416 | 0.443 | 0.410 | 0.427 | 0.385 | 0.447 | **0.354** | **0.414** | 0.519 | 0.500 | 0.461 | 0.454 | 0.613 | 0.539 | 1.913 | 1.159 | 1.550 | 0.998 | 1.402 | 0.968 | 1.280 | 0.932 |
| ILI | 2.139 | 0.931 | 2.497 | 1.004 | 7.382 | 2.003 | 2.616 | 1.090 | 2.847 | 1.144 | **2.077** | **0.914** | 3.006 | 1.161 | 7.635 | 2.050 | 5.137 | 1.544 | 4.839 | 1.485 | 4.724 | 1.445 |

Table 3: Short-term forecasting task on M4. The prediction lengths are in [6, 48] and results are weighted averaged from several datasets under different sample intervals. See Table 14 for full results.

| Models | TimesNet (Ours) | N-HiTS (2022) | N-BEATS (2019) | ETSformer (2022) | LightTS (2022) | DLinear (2023) | FEDformer (2022) | Stationary (2022a) | Autoformer (2021) | Pyraformer (2021a) | Informer (2021) | LogTrans (2019) | Reformer (2020) |
|---|---|---|---|---|---|---|---|---|---|---|---|---|---|
| SMAPE | **11.829** | 11.927 | 11.851 | 14.718 | 13.525 | 13.639 | 12.840 | 12.780 | 12.909 | 16.987 | 14.086 | 16.018 | 18.200 |
| MASE | **1.585** | 1.613 | 1.599 | 2.408 | 2.111 | 2.095 | 1.701 | 1.756 | 1.771 | 3.265 | 2.718 | 3.010 | 4.223 |
| OWA | **0.851** | 0.861 | 0.855 | 1.172 | 1.051 | 1.051 | 0.918 | 0.930 | 0.939 | 1.480 | 1.230 | 1.378 | 1.775 |

**Results** TimesNet shows great performance in both long-term and short-term settings (Table 2–3). Concretely, TimesNet achieves state-of-the-art in more than 80% of cases in long-term forecasting (Table 13). For the M4 dataset, since the time series are collected from different sources, the temporal variations can be quite diverse, making forecasting much more challenging. Our model still performs best in this task, surpassing extensive advanced MLP-based and Transformer-based models.

## 4.3 IMPUTATION

**Setups** Real-world systems always work continuously and are monitored by automatic observation equipment. However, due to malfunctions, the collected time series can be partially missing, making the downstream analysis difficult. Thus, imputation is widely-used in practical applications. In this paper, we select the datasets from the electricity and weather scenarios as our benchmarks, including ETT (Zhou et al., 2021), Electricity (UCI) and Weather (Wetterstation), where the data-missing problem happens commonly. To compare the model capacity under different proportions of missing data, we randomly mask the time points in the ratio of {12.5%, 25%, 37.5%, 50%}.

**Results** Due to the missing time points, the imputation task requires the model to discover underlying temporal patterns from the irregular and partially observed time series. As shown in Table 4, our proposed TimesNet still achieves the consistent state-of-the-art in this difficult task, verifying the model capacity in capturing temporal variation from extremely complicated time series.

## 4.4 CLASSIFICATION

**Setups** Time series classification can be used in recognition and medical diagnosis (Moody et al., 2011). We adopt the sequence-level classification to verify the model capacity in high-level representation learning. Concretely, we select 10 multivariate datasets from UEA Time Series Classification Archive (Bagnall et al., 2018), covering the gesture, action and audio recognition, medical diagnosis by heartbeat monitoring and other practical tasks. Then, we pre-process the datasets following the descriptions in (Zerveas et al., 2021), where different subsets have different sequence lengths.

Table 4: Imputation task. We randomly mask $\{12.5\%, 25\%, 37.5\%, 50\%\}$ time points in length-96 time series. The results are averaged from 4 different mask ratios. See Table 16 for full results.

| Models | TimesNet (Ours) | | ETSformer (2022) | | LightTS (2022) | | DLinear (2023) | | FEDformer (2022) | | Stationary (2022a) | | Autoformer (2021) | | Pyraformer (2021a) | | Informer (2021) | | LogTrans (2019) | | Reformer (2020) | |
|---|---|---|---|---|---|---|---|---|---|---|---|---|---|---|---|---|---|---|---|---|---|---|---|
| Metric | MSE | MAE | MSE | MAE | MSE | MAE | MSE | MAE | MSE | MAE | MSE | MAE | MSE | MAE | MSE | MAE | MSE | MAE | MSE | MAE | MSE | MAE |
| ETTm1 | **0.027** | **0.107** | 0.120 | 0.253 | 0.104 | 0.218 | 0.093 | 0.206 | 0.062 | 0.177 | 0.036 | 0.126 | 0.051 | 0.150 | 0.717 | 0.570 | 0.071 | 0.188 | 0.050 | 0.154 | 0.055 | 0.166 |
| ETTm2 | **0.022** | **0.088** | 0.208 | 0.327 | 0.046 | 0.151 | 0.096 | 0.208 | 0.101 | 0.215 | 0.026 | 0.099 | 0.029 | 0.105 | 0.465 | 0.508 | 0.156 | 0.292 | 0.119 | 0.246 | 0.157 | 0.280 |
| ETTh1 | **0.078** | **0.187** | 0.202 | 0.329 | 0.284 | 0.373 | 0.201 | 0.306 | 0.117 | 0.246 | 0.094 | 0.201 | 0.103 | 0.214 | 0.842 | 0.682 | 0.161 | 0.279 | 0.219 | 0.332 | 0.122 | 0.245 |
| ETTh2 | **0.049** | **0.146** | 0.367 | 0.436 | 0.119 | 0.250 | 0.142 | 0.259 | 0.163 | 0.279 | 0.053 | 0.152 | 0.055 | 0.156 | 1.079 | 0.792 | 0.337 | 0.452 | 0.186 | 0.318 | 0.234 | 0.352 |
| Electricity | **0.092** | **0.210** | 0.214 | 0.339 | 0.131 | 0.262 | 0.132 | 0.260 | 0.130 | 0.259 | 0.100 | 0.218 | 0.101 | 0.225 | 0.297 | 0.382 | 0.222 | 0.328 | 0.175 | 0.303 | 0.200 | 0.313 |
| Weather | **0.030** | **0.054** | 0.076 | 0.171 | 0.055 | 0.117 | 0.052 | 0.110 | 0.099 | 0.203 | 0.032 | 0.059 | 0.031 | 0.057 | 0.152 | 0.235 | 0.045 | 0.104 | 0.039 | 0.076 | 0.038 | 0.087 |

**Results** As shown in Figure 5, TimesNet achieves the best performance with an average accuracy of 73.6%, surpassing the previous state-of-the-art classical method Rocket (72.5%) and deep model Flowformer (73.0%). It is also notable that the MLP-based model DLinear fails in this classification task (67.5%), which performs well in some time series forecasting datasets. This is because DLinear only adopts a one-layer MLP model on the temporal dimension, which might be suitable for some autoregressive tasks with fixed temporal

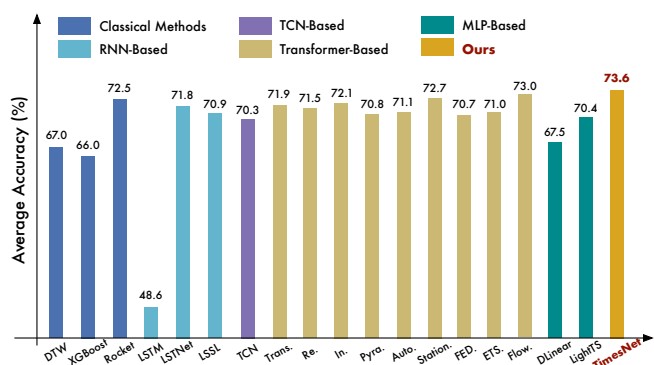

Figure 5: Model comparison in classification. "∗." in the Transformers indicates the name of ∗former. The results are averaged from 10 subsets of UEA. See Table 17 in Appendix for full results.

dependencies but will degenerate a lot in learning high-level representations. In contrast, TimesNet unifies the temporal 2D-variation in 2D space, which is convenient to learn informative representation by 2D kernels, thereby benefiting the classification task that requires hierarchical representations.

## 4.5 ANOMALY DETECTION

**Setups** Detecting anomalies from monitoring data is vital to industrial maintenance. Since the anomalies are usually hidden in the large-scale data, making the data labeling hard, we focus on unsupervised time series anomaly detection, which is to detect the abnormal time points. We compare models on five widely-used anomaly detection benchmarks: SMD (Su et al., 2019), MSL (Hundman et al., 2018), SMAP (Hundman et al., 2018), SWaT (Mathur & Tippenhauer, 2016), PSM (Abdulaal et al., 2021), covering service monitoring, space & earth exploration, and water treatment applications. Following the pre-processing methods in Anomaly Transformer (2021), we split the dataset into consecutive non-overlapping segments by sliding window. In previous works, the reconstruction is a classical task for unsupervised point-wise representation learning, where the reconstruction error is a natural anomaly criterion. For a fair comparison, we only change the base models for reconstruction and use the classical reconstruction error as the shared anomaly criterion for all experiments.

**Results** Table 5 demonstrates that TimesNet still achieves the best performance in anomaly detection, outperforming the advanced Transformer-based models FEDformer (2022) and Autoformer (2021). The canonical Transformer performs worse in this task (averaged F1-score 76.88%). This may come from that anomaly detection requires the model to find out the rare abnormal temporal patterns (Lai et al., 2021), while the vanilla attention mechanism calculates the similarity between each pair of time points, which can be distracted by the dominant normal time points. Besides, by taking the periodicity into consideration, TimesNet, FEDformer and Autoformer all achieve great performance. Thus, these results also demonstrate the importance of periodicity analysis, which can highlight variations that violate the periodicity implicitly, further benefiting the anomaly detection.

Table 5: Anomaly detection task. We calculate the F1-score (as %) for each dataset. *. means the *former. A higher value of F1-score indicates a better performance. See Table 15 for full results.

| Models | TimesNet (ResNeXt) | TimesNet (Inception) | ETS. (2022) | FED. (2022) | LightTS (2022) | DLinear (2023) | Stationary (2022a) | Auto. (2021) | Pyra. (2021a) | Anomaly* (2021) | In. (2021) | Re. (2020) | LogTrans (2019) | Trans. (2017) |
|---|---|---|---|---|---|---|---|---|---|---|---|---|---|---|
| SMD | **85.81** | 85.12 | 83.13 | 85.08 | 82.53 | 77.10 | 84.72 | 85.11 | 83.04 | 85.49 | 81.65 | 75.32 | 76.21 | 79.56 |
| MSL | **85.15** | 84.18 | 85.03 | 78.57 | 78.95 | 84.88 | 77.50 | 79.05 | 84.86 | 83.31 | 84.06 | 84.40 | 79.57 | 78.68 |
| SMAP | **71.52** | 70.85 | 69.50 | 70.76 | 69.21 | 69.26 | 71.09 | 71.12 | 71.09 | 71.18 | 69.92 | 70.40 | 69.97 | 69.70 |
| SWaT | 91.74 | 92.10 | 84.91 | 93.19 | **93.33** | 87.52 | 79.88 | 92.74 | 91.78 | 83.10 | 81.43 | 82.80 | 80.52 | 80.37 |
| PSM | **97.47** | 95.21 | 91.76 | 97.23 | 97.15 | 93.55 | 97.29 | 93.29 | 82.08 | 79.40 | 77.10 | 73.61 | 76.74 | 76.07 |
| Avg F1 | **86.34** | 85.49 | 82.87 | 84.97 | 84.23 | 82.46 | 82.08 | 84.26 | 82.57 | 80.50 | 78.83 | 77.31 | 76.60 | 76.88 |

∗ We replace the joint criterion in Anomaly Transformer (2021) with reconstruction error for fair comparison.

## 4.6 MODEL ANALYSIS

**Representation analysis**   We attempt to explain model performance from the representation learning aspect. From Figure 6, we can find that the better performance in forecasting and anomaly detection corresponds to the higher CKA similarity (2019), which is opposite to the imputation and classification tasks. Note that the lower CKA similarity means that the representations are distinguishing among different layers, namely hierarchical representations. Thus, these results also indicate the property of representations that each task requires. As shown in Figure 6, TimesNet can learn appropriate representations for different tasks, such as low-level representations for forecasting and reconstruction in anomaly detection, and hierarchical representations for imputation and classification. In contrast, FEDformer (2022) performs well in forecasting and anomaly detection tasks but fails in learning hierarchical representations, resulting in poor performance in imputation and classification. These results also verify the task-generality of our proposed TimesNet as a foundation model.

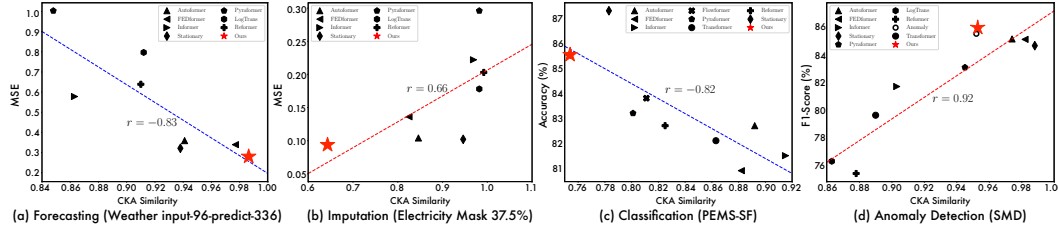

(a) Forecasting (Weather input-96-predict-336)  (b) Imputation (Electricity Mask 37.5%)  (c) Classification (PEMS-SF)  (d) Anomaly Detection (SMD)

Figure 6: Representation analysis in four tasks. For each model, we calculate the centered kernel alignment (CKA) similarity (2019) between representations from the first and the last layers. A higher CKA similarity indicates more similar representations. TimesNet is marked by red stars.

**Temporal 2D-variations**   We provide a case study of temporal 2D-variations in Figure 7. We can find that TimesNet can capture the multi-periodicities precisely. Besides, the transformed 2D tensor is highly structured and informative, where the columns and rows can reflect the localities between time points and periods respectively, supporting our motivation in adopting 2D kernels for representation learning. See Appendix D for more visualizations.

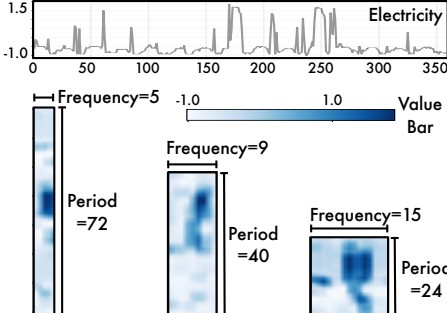

Figure 7: A case of temporal 2D-variations.

## 5 CONCLUSION AND FUTURE WORK

This paper presents the TimesNet as a task-general foundation model for time series analysis. Motivated by the multi-periodicity, TimesNet can ravel out intricate temporal variations by a modular architecture and capture intraperiod- and interperiod-variations in 2D space by a parameter-efficient inception block. Experimentally, TimesNet shows great generality and performance in five mainstream analysis tasks. In the future, we will further explore large-scale pre-training methods in time series, which utilize TimesNet as the backbone and can generally benefit extensive downstream tasks.

## ACKNOWLEDGMENTS

This work was supported by the National Key Research and Development Plan (2020AAA0109201), National Natural Science Foundation of China (62022050 and 62021002), Civil Aircraft Research Project (MZJ3-2N21), Beijing Nova Program (Z201100006820041), and CCF-Ant Group Green Computing Fund.

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

## A  IMPLEMENTATION DETAILS

We provide the dataset descriptions and experiment configurations in Table 6 and 7. All experiments are repeated three times, implemented in PyTorch (Paszke et al., 2019) and conducted on a single NVIDIA TITAN RTX 24GB GPU. To make the model handle various dimensions of input series from different datasets, we select the $d_{\text{model}}$ based on the input series dimension $C$ by $\min\{\max\{2^{\lceil \log C \rceil}, d_{\text{min}}\}, d_{\text{max}}\}$ (see Table 7 for details of $d_{\text{min}}$ and $d_{\text{max}}$). This protocol can make the model powerful enough for multiple variates and also keep the model size compact.

All the baselines that we reproduced are implemented based on configurations of the original paper or official code. It is also notable that none of the previous methods are proposed for general time series analysis. For a fair comparison, we keep the input embedding and the final projection layer the same among different base models and only evaluate the capability of base models. Especially for the forecasting task, we use a MLP on temporal dimension to get the initialization of predicted future. Since we focus on the temporal variation modeling, we also adopt the Series Stationarization from Non-stationary Transformer (Liu et al., 2022a) to eliminate the affect the distribution shift.

For the metrics, we adopt the mean square error (MSE) and mean absolute error (MAE) for long-term forecasting and imputations. For anomaly detection, we adopt the F1-score, which is the harmonic mean of precision and recall. For the short-term forecasting, following the N-BEATS (Oreshkin et al., 2019), we adopt the symmetric mean absolute percentage error (SMAPE), mean absolute scaled error (MASE) and overall weighted average (OWA) as the metrics, where OWA is a special metric used in M4 competition. These metrics can be calculated as follows:

$$\text{SMAPE} = \frac{200}{H} \sum_{i=1}^{H} \frac{|\mathbf{X}_i - \widehat{\mathbf{X}}_i|}{|\mathbf{X}_i| + |\widehat{\mathbf{X}}_i|}, \qquad \text{MAPE} = \frac{100}{H} \sum_{i=1}^{H} \frac{|\mathbf{X}_i - \widehat{\mathbf{X}}_i|}{|\mathbf{X}_i|},$$

$$\text{MASE} = \frac{1}{H} \sum_{i=1}^{H} \frac{|\mathbf{X}_i - \widehat{\mathbf{X}}_i|}{\frac{1}{H-m} \sum_{j=m+1}^{H} |\mathbf{X}_j - \mathbf{X}_{j-m}|}, \qquad \text{OWA} = \frac{1}{2} \left[ \frac{\text{SMAPE}}{\text{SMAPE}_{\text{Naïve2}}} + \frac{\text{MASE}}{\text{MASE}_{\text{Naïve2}}} \right],$$

where $m$ is the periodicity of the data. $\mathbf{X}, \widehat{\mathbf{X}} \in \mathbb{R}^{H \times C}$ are the ground truth and prediction results of the future with $H$ time pints and $C$ dimensions. $\mathbf{X}_i$ means the $i$-th future time point.

## B    HYPER-PARAMETER SENSITIVITY

We introduce a hyper-parameter in Equation 1 to select the most significant frequencies. We provide the sensitivity analysis for these two hyper-parameters in Figure 8. We can find that our proposed TimesNet can present a stable performance under different choices of $k$ in all four tasks.

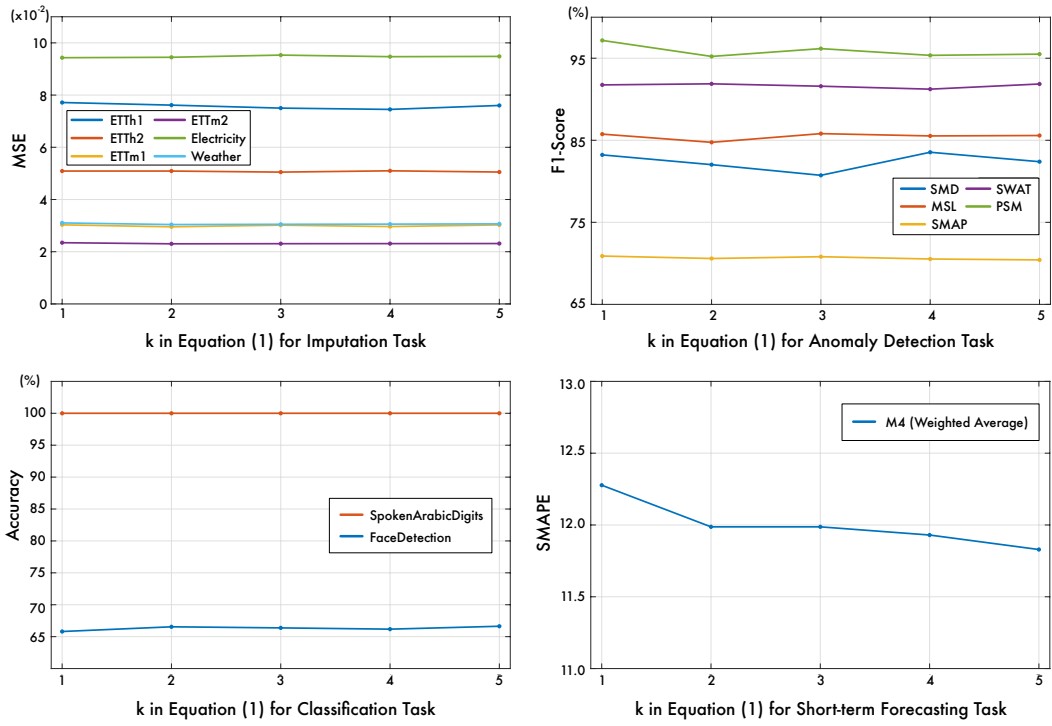

Figure 8: Sensitivity analysis of hyper-parameters $k$ in imputation and anomaly detection task. Especially, we select the 37.5% mask ratio for hyper-parameter experiments. For classification, we choose the two largest subsets: SpokenArobicDigits and FaceDetection for evaluation. For short-term forecasting, we adopt the weighted average for sensitivity analysis.

Besides, from Figure 8, we can also find out the following observations:

- For the low-level modeling tasks, such as forecasting and anomaly detection, the selection of $k$ will affect the final performance more. This may come from that $k$ will directly affect the amount of information in deep representations.

Table 6: Dataset descriptions. The dataset size is organized in (Train, Validation, Test).

| Tasks | Dataset | Dim | Series Length | Dataset Size | Information (Frequency) |
|---|---|---|---|---|---|
| Forecasting (Long-term) | ETTm1, ETTm2 | 7 | {96, 192, 336, 720} | (34465, 11521, 11521) | Electricity (15 mins) |
| | ETTh1, ETTh2 | 7 | {96, 192, 336, 720} | (8545, 2881, 2881) | Electricity (15 mins) |
| | Electricity | 321 | {96, 192, 336, 720} | (18317, 2633, 5261) | Electricity (Hourly) |
| | Traffic | 862 | {96, 192, 336, 720} | (12185, 1757, 3509) | Transportation (Hourly) |
| | Weather | 21 | {96, 192, 336, 720} | (36792, 5271, 10540) | Weather (10 mins) |
| | Exchange | 8 | {96, 192, 336, 720} | (5120, 665, 1422) | Exchange rate (Daily) |
| | ILI | 7 | {24, 36, 48, 60} | (617, 74, 170) | Illness (Weekly) |
| Forecasting (short-term) | M4-Yearly | 1 | 6 | (23000, 0, 23000) | Demographic |
| | M4-Quarterly | 1 | 8 | (24000, 0, 24000) | Finance |
| | M4-Monthly | 1 | 18 | (48000, 0, 48000) | Industry |
| | M4-Weakly | 1 | 13 | (359, 0, 359) | Macro |
| | M4-Daily | 1 | 14 | (4227, 0, 4227) | Micro |
| | M4-Hourly | 1 | 48 | (414, 0, 414) | Other |
| Imputation | ETTm1, ETTm2 | 7 | 96 | (34465, 11521, 11521) | Electricity (15 mins) |
| | ETTh1, ETTh2 | 7 | 96 | (8545, 2881, 2881) | Electricity (15 mins) |
| | Electricity | 321 | 96 | (18317, 2633, 5261) | Electricity (15 mins) |
| | Weather | 21 | 96 | (36792, 5271, 10540) | Weather (10 mins) |
| Classification (UEA) | EthanolConcentration | 3 | 1751 | (261, 0, 263) | Alcohol Industry |
| | FaceDetection | 144 | 62 | (5890, 0, 3524) | Face (250Hz) |
| | Handwriting | 3 | 152 | (150, 0, 850) | Handwriting |
| | Heartbeat | 61 | 405 | (204, 0, 205) | Heart Beat |
| | JapaneseVowels | 12 | 29 | (270, 0, 370) | Voice |
| | PEMS-SF | 963 | 144 | (267, 0, 173) | Transportation (Daily) |
| | SelfRegulationSCP1 | 6 | 896 | (268, 0, 293) | Health (256Hz) |
| | SelfRegulationSCP2 | 7 | 1152 | (200, 0, 180) | Health (256Hz) |
| | SpokenArabicDigits | 13 | 93 | (6599, 0, 2199) | Voice (11025Hz) |
| | UWaveGestureLibrary | 3 | 315 | (120, 0, 320) | Gesture |
| Anomaly Detection | SMD | 38 | 100 | (566724, 141681, 708420) | Server Machine |
| | MSL | 55 | 100 | (44653, 11664, 73729) | Spacecraft |
| | SMAP | 25 | 100 | (108146, 27037, 427617) | Spacecraft |
| | SWaT | 51 | 100 | (396000, 99000, 449919) | Infrastructure |
| | PSM | 25 | 100 | (105984, 26497, 87841) | Server Machine |

Table 7: Experiment configuration of TimesNet. All the experiments use the ADAM (2015) optimizer with the default hyperparameter configuration for $(\beta_1, \beta_2)$ as (0.9, 0.999).

| Tasks / Configurations | Model Hyper-parameter | | | | Training Process | | | |
|---|---|---|---|---|---|---|---|---|
| | $k$ (Equ. 1) | Layers | $d_{\min}$ † | $d_{\max}$ † | LR* | Loss | Batch Size | Epochs |
| Long-term Forecasting | 5 | 2 | 32 | 512 | $10^{-4}$ | MSE | 32 | 10 |
| Short-term Forecasting | 5 | 2 | 16 | 64 | $10^{-3}$ | SMAPE | 16 | 10 |
| Imputation | 3 | 2 | 64 | 128 | $10^{-3}$ | MSE | 16 | 10 |
| Classification | 3 | 2 | 32 | 64 | $10^{-3}$ | Cross Entropy | 16 | 30 |
| Anomaly Detection | 3 | 3 | 32 | 128 | $10^{-4}$ | MSE | 128 | 10 |

† $d_{\text{model}} = \min\{\max\{2^{\lceil \log C \rceil}, d_{\min}\}, d_{\max}\}$, where $C$ is input series dimension.
∗ LR means the initial learning rate.

- For the high-level modeling tasks, such as classification and imputation, the model performance will be more robust to the selection of $k$, since the key to these tasks is to extract hierarchical representations.

Giving consideration to both efficiency and performance, we set $k = 3$ for imputation, classification and anomaly detection and $k = 5$ for short-term forecasting.

## C  ABLATION STUDIES

To elaborate the property of our proposed TimesNet, we conduct detailed ablations on the representation leaning in 2D space, model architecture and adaptive aggregation.

**2D space**  As shown in Table 8, replacing the inception block with more powerful blocks will bring further performance promotion, such as ResNeXt (Xie et al., 2017), Swin Transformer (Liu et al., 2021b) and ConvNeXt (Liu et al., 2022b). It is also notable that using the independent parameters will also bring improvement, while this will cause the model size related to the selection of hyperparameter $k$. Considering the efficiency and model performance, we choose the parameter-efficient inception block as our final solution. These results also verify that our design bridges the 1D time series analysis with 2D computer vision backbones.

Table 8: Ablations on the representation leaning in 2D space, where we replace the parameter-efficient inception with other well-acknowledged vision backbones. See Figure 4 for efficiency comparison.

| Datasets | SMD | | | MSL | | | SMAP | | | SWaT | | | PSM | | | Avg F1 |
|---|---|---|---|---|---|---|---|---|---|---|---|---|---|---|---|---|
| Metrics | P | R | F1 | P | R | F1 | P | R | F1 | P | R | F1 | P | R | F1 | (%) |
| ResNet | 87.64 | 80.33 | 83.82 | 83.99 | 85.42 | 84.70 | 92.09 | 57.90 | 71.10 | 86.59 | 95.81 | 90.97 | 98.24 | 89.12 | 93.45 | 84.81 |
| ResNeXt | 88.66 | 83.14 | 85.81 | 83.92 | 86.42 | 85.15 | 92.52 | 58.29 | 71.52 | 86.76 | 97.32 | 91.74 | 98.19 | 96.76 | **97.47** | 86.34 |
| Swin Transformer | 88.51 | 82.22 | 85.25 | 87.36 | 86.93 | 87.14 | 92.12 | 57.60 | 70.88 | 89.02 | 95.81 | 92.29 | 98.45 | 91.80 | 95.01 | 86.11 |
| ConvNext | 87.89 | 84.67 | **86.25** | 87.31 | 86.93 | 87.12 | 92.42 | 59.19 | **72.16** | 89.05 | 95.81 | 92.31 | 97.99 | 95.28 | 96.62 | **86.89** |
| Inception (Ind*) | 87.54 | 81.04 | 84.17 | 87.44 | 86.93 | **87.18** | 91.92 | 57.69 | 70.89 | 88.98 | 96.00 | **92.36** | 98.11 | 89.13 | 93.41 | 85.60 |
| Inception (Shared*) | 87.76 | 82.63 | 85.12 | 82.97 | 85.42 | 84.18 | 91.50 | 57.80 | 70.85 | 88.31 | 96.24 | 92.10 | 98.22 | 92.21 | 95.21 | 85.49 |

∗ In this paper, we adopt a parameter-efficient design that uses the same parameters for $k$ different transformed 2D tensors, namely *Shared*. For comparison, we also compare with the independent design, that uses different parameters for different transformed 2D tensors, namely *Ind*.

**Model architecture**  We also conduct experiments on different architectures. Surprisingly, as shown in Table 9, we find that combining with the deep decomposition architecture in Autoformer (Wu et al., 2021) cannot bring further promotion. These results may come from that in the case that the input series already present clear periodicity, our design can capture the 2D-variations effectively. As for the case that they are without clear periodicity, the model will learn the most significant frequency as 1, where the trend of time series be covered by the intraperiod-variation modeling. These results also verify that our proposed TimesNet can handle the analysis for time series without clear periodicity.

Besides, in this paper, to take advantages of the deep representations, we place the transformation from 1D-variations to 2D-variations in every TimesBlock. Here, we compare our design with the case that only conducts the transformation on the raw input series. From Table 9, we can find that the performance of *Transform raw data* degenerates a lot (Avg F1-score: 85.49% → 84.85%), indicating the advantages of our design.

Table 9: Ablations on model architecture. + *decomposition* is to combine the deep decomposition architecture proposed by Autoformer (Wu et al., 2021) with TimesNet. *Transform raw data* refers conducting the transformation on the original time series, instead of deep features.

| Datasets | SMD | | | MSL | | | SMAP | | | SWaT | | | PSM | | | Avg F1 |
|---|---|---|---|---|---|---|---|---|---|---|---|---|---|---|---|---|
| Metrics | P | R | F1 | P | R | F1 | P | R | F1 | P | R | F1 | P | R | F1 | (%) |
| TimesNet | 87.76 | 82.63 | **85.12** | 82.97 | 85.42 | 84.18 | 91.50 | 57.80 | **70.85** | 88.31 | 96.24 | 92.10 | 98.22 | 92.21 | 95.21 | **85.49** |
| + decomposition | 87.44 | 78.49 | 82.72 | 83.48 | 86.47 | 84.95 | 91.64 | 57.34 | 70.54 | 89.68 | 95.60 | **92.54** | 98.42 | 93.12 | **95.69** | 85.29 |
| Transform raw data | 86.83 | 79.17 | 82.82 | 85.23 | 86.47 | **85.84** | 91.92 | 57.60 | 70.82 | 87.68 | 95.81 | 91.57 | 97.64 | 89.14 | 93.20 | 84.85 |

**Adaptive aggregation** As shown in Equation 6, following the design in Autoformer (2021), we adopt the amplitudes after the Softmax function as aggregation weights of processed tensors $\{\widehat{\mathbf{X}}_{1D}^{l,1}, \cdots, \widehat{\mathbf{X}}_{1D}^{l,k}\}$. Here we include two variants for comparison. The first is directly-sum, namely $\sum_{i=1}^{k} \widehat{\mathbf{X}}_{1D}^{l,i}$. The second is to remove the Softmax function, that is $\sum_{i=1}^{k} \mathbf{A}_{f_i}^{l-1} \times \widehat{\mathbf{X}}_{1D}^{l,i}$. Table 10 demonstrates that our design in adaptive aggregation performs the best.

Table 10: Ablations on adaptive aggregation.

| Datasets | SMD | | | MSL | | | SMAP | | | SWaT | | | PSM | | | Avg F1 |
|---|---|---|---|---|---|---|---|---|---|---|---|---|---|---|---|---|
| Metrics | P | R | F1 | P | R | F1 | P | R | F1 | P | R | F1 | P | R | F1 | (%) |
| TimesNet | 87.76 | 82.63 | 85.12 | 82.97 | 85.42 | 84.18 | 91.50 | 57.80 | 70.85 | 88.31 | 96.24 | 92.10 | 98.22 | 92.21 | 95.21 | **85.49** |
| Directly-sum | 87.10 | 78.93 | 82.81 | 85.81 | 85.42 | 85.62 | 91.35 | 57.58 | 70.64 | 87.28 | 96.00 | 91.43 | 98.13 | 87.22 | 92.35 | 84.57 |
| Removing-Softmax | 87.27 | 79.31 | 83.10 | 83.91 | 86.47 | 85.17 | 91.93 | 58.57 | 71.55 | 87.13 | 95.81 | 91.27 | 98.00 | 92.48 | 95.16 | 85.25 |

## D  MORE REPRESENTATION ANALYSIS

To give an intuitive understanding of 2D-variations, we visualize the transformed 2D tensors in Figure 9. From the visualization, we can obtain the following observations:

- The interperiod-variation can present the long-term trends of time series. For example, in the first case of Exchange, the values in each row decrease from left to right, indicating the downtrend of the raw series. And for the ETTh1 dataset, the values in each row are similar to each other, reflecting the global stable variation of the raw series.

- For the time series without clear periodicity, the temporal 2D-variations can still present informative 2D structure. If the frequency is one, the intraperiod-variation is just the original variation of raw series. Besides, the interperiod-variation can also present the long-term trend, benefiting the temporal variation modeling.

- The transformed 2D-variations demonstrate two types of localities. Firstly, for each column (intraperiod-variation), the adjacent values are close to each other, presenting the locality among adjacent time points. Secondly, for each row (interperiod-variation), the adjacent values are also close, corresponding to the locality among adjacent periods. Note that the non-adjacent periods can be quite different from each other, which can be caused by global trend, such as the case from the Exchange dataset. These observations of localities also motivate us to adopt the 2D kernel for representation learning.

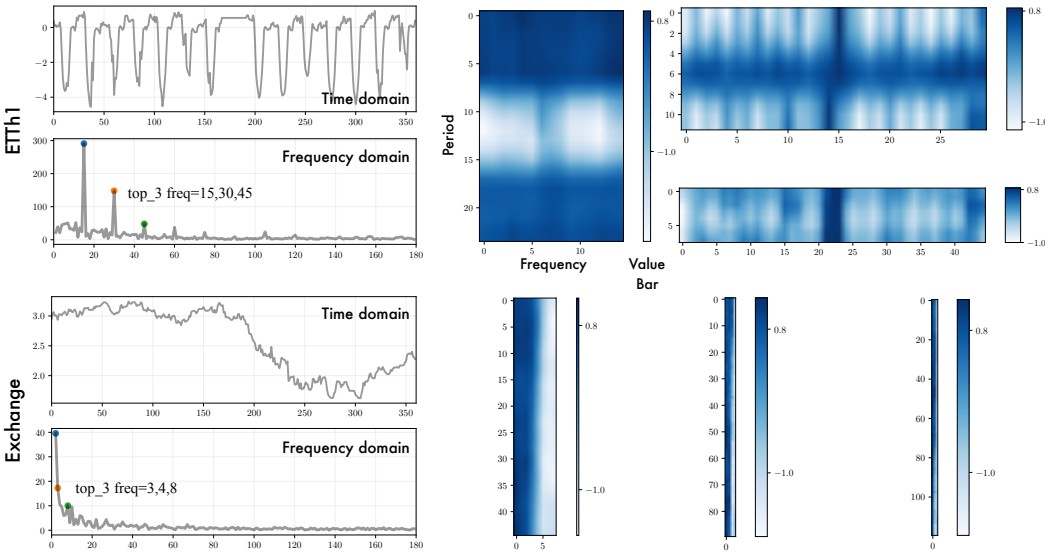

Figure 9: More showcases for temporal 2D-variations.

# E    MULTI-PERIODICITY OF TIME SERIES

As shown in Figure 10, we calculate the density of each period length for different datasets. We can find that real-world time series present multi-periodicity to some extent. For example, the Electricity dataset contains the periods with length-12 and length-24.

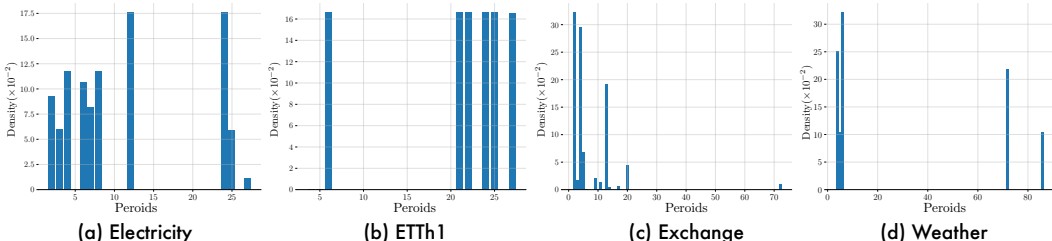

Figure 10: Statistics of period length in experimental datasets. We conduct FFT to the raw data and select the top-6 significant frequencies for each length-96 segment. Then, we record the corresponding period lengths and plot the normalized density for each period length.

# F    SHOWCASES

To provide a clear comparison among different models, we provide showcases to the regression tasks, including the imputation (Figure 11), long-term forecasting (Figure 12) and short-term forecasting (Figure 13). Especially in the imputation task, the MLP-based models degenerate a lot. This is because the input series has been randomly masked. However, the MLP-based models adopt the fixed model parameter to model the temporal dependencies among time points, thereby failing in this task.

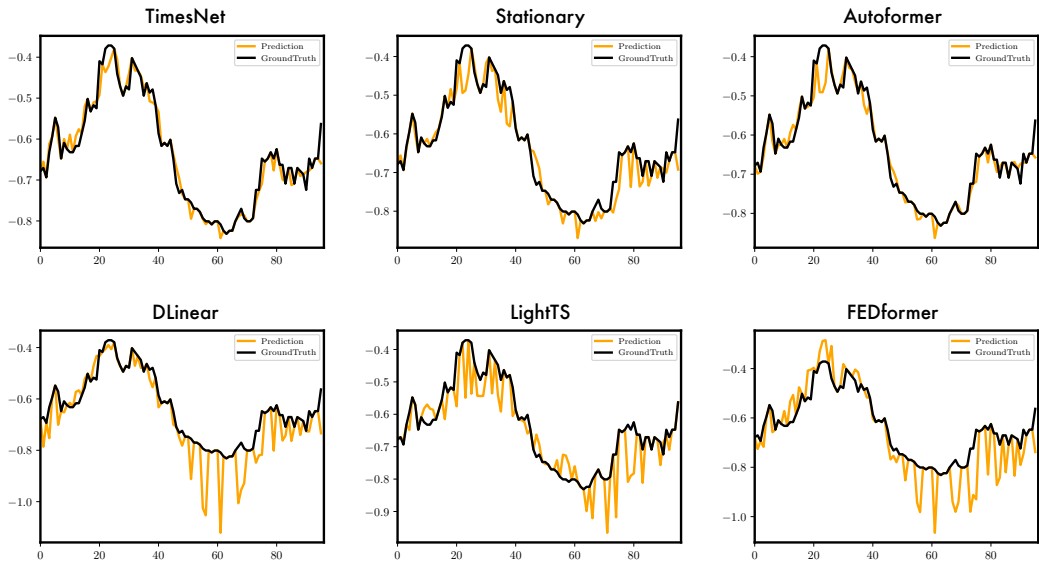

Figure 11: Visualization of ETTm1 imputation results given by models under the 50% mask ratio setting. The black lines stand for the ground truth and the orange lines stand for predicted values.

# G    MODEL EFFICIENCY ANALYSIS

To summarize the model performance and efficiency, we calculate the relative performance rankings for comparing baselines. The rankings are compared from the common models used in all five tasks: LSTM (1997) and LSSL (2022); TCN (2019); LightTS (2022) and DLinear (2023); Reformer (2020), Informer (2021), Pyraformer (2021a), Autoformer (2021), FEDformer (2022), Non-stationary Transformer (2022a), ETSformer (2022) and our proposed TimesNet, namely 13 models in total.

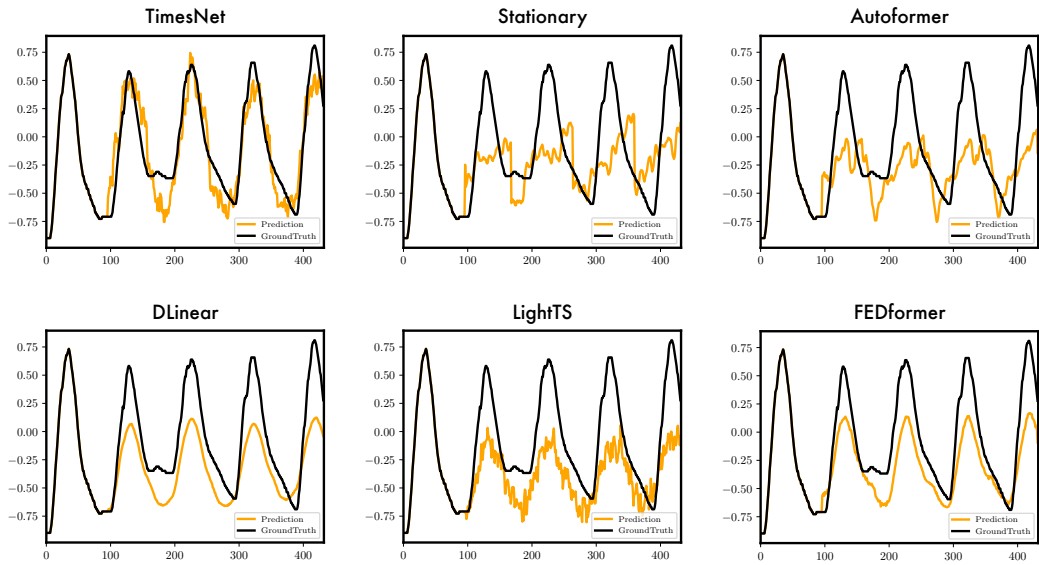

Figure 12: Visualization of ETTm2 predictions by different models under the input-96-predict-336 setting. The black lines stand for the ground truth and the orange lines stand for predicted values.

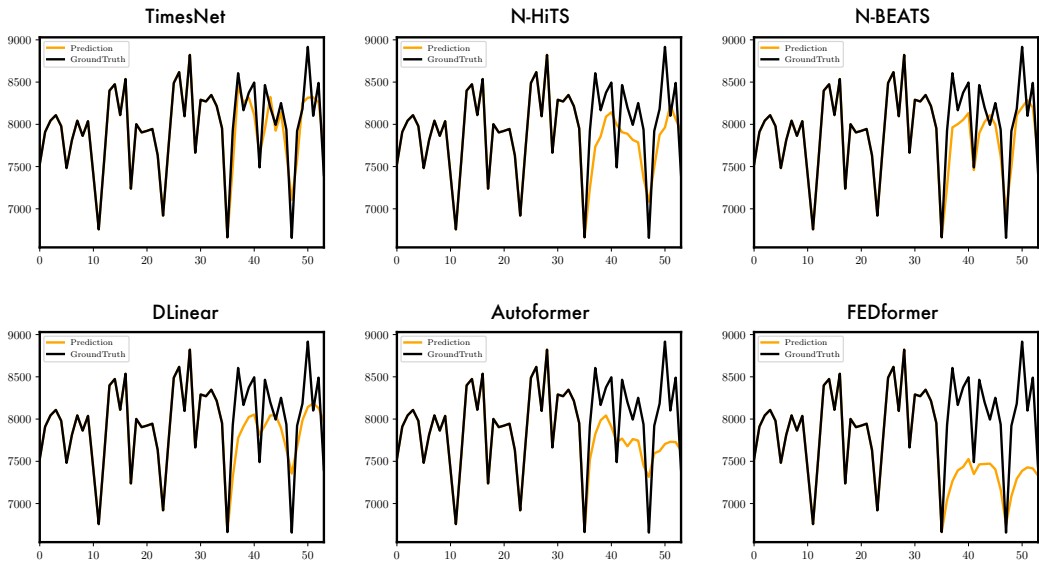

Figure 13: Visualization of M4 predictions by different models. The black lines stand for the ground truth and the orange lines stand for predicted values.

As shown in Table 11, our proposed TimesNet achieves the best performance in all five tasks. Among the top three models, TimesNet also achieves the greatest efficiency. Compared to MLP-based models, our proposed TimesNet shows a significant advantage in performance. And benefiting from the utilization of 2D kernels and parameter-efficient design, the parameter size is invariant when the input series changes. Compared to Transformer-based models, TimesNet is with great efficiency in GPU memory, which is essential in long sequence modeling.

Table 11: Model efficiency comparison and their rankings in five tasks. The efficiency measurements are recorded on the imputation task of ETTh1 dataset. The rankings are organized in the order of long- and short-term forecasting, imputation, classification and anomaly detection. "/" indicates the out-of-memory situation. A smaller ranking means better performance.

| Models | | Parameter | GPU Memory | Running Time | Ranking | |
|---|---|---|---|---|---|---|
| Series Length | | (MB) | (MiB) | (s / iter) | Five tasks | Avg Ranking |
| **TimesNet (ours)** | 384 | 0.067 | 1245 | 0.024 | (1, 1, 1, 1, 1) | 1.0 |
| | 768 | 0.067 | 1585 | 0.040 | | |
| | 1536 | 0.067 | 2491 | 0.045 | | |
| | 3072 | 0.067 | 2353 | 0.073 | | |
| Non-stationary Transformer | 384 | 1.884 | 2321 | 0.046 | (3, 2, 2, 2, 8) | 3.4 |
| | 768 | 1.910 | 4927 | 0.118 | | |
| | 1536 | 1.961 | / | / | | |
| | 3072 | / | / | / | | |
| Autoformer | 384 | 1.848 | 2101 | 0.070 | (7, 4, 3, 5, 3) | 4.4 |
| | 768 | 1.848 | 3209 | 0.071 | | |
| | 1536 | 1.848 | 5395 | 0.129 | | |
| | 3072 | 1.848 | 10043 | 0.255 | | |
| FEDformer | 384 | 2.901 | 5977 | 0.807 | (4, 3, 6, 9, 2) | 4.8 |
| | 768 | 2.901 | 7111 | 1.055 | | |
| | 1536 | 2.901 | 9173 | 1.482 | | |
| | 3072 | 2.901 | / | / | | |
| LightTS | 384 | 0.163 | 1055 | 0.009 | (6, 5, 4, 10, 4) | 5.8 |
| | 768 | 0.614 | 1077 | 0.013 | | |
| | 1536 | 2.403 | 1127 | 0.015 | | |
| | 3072 | 9.534 | 1311 | 0.030 | | |
| DLinear | 384 | 0.296 | 1057 | 0.006 | (2, 6, 5, 12, 7) | 6.4 |
| | 768 | 1.181 | 1093 | 0.006 | | |
| | 1536 | 4.722 | 1159 | 0.007 | | |
| | 3072 | 18.881 | 1433 | 0.026 | | |
| ETSformer | 384 | 1.123 | 1831 | 0.042 | (5, 9, 9, 6, 5) | 6.8 |
| | 768 | 1.123 | 2565 | 0.047 | | |
| | 1536 | 1.123 | 4081 | 0.072 | | |
| | 3072 | 1.123 | 7065 | 0.143 | | |
| Informer | 384 | 1.903 | 1577 | 0.044 | (10, 8, 8, 3, 9) | 7.6 |
| | 768 | 1.903 | 2125 | 0.047 | | |
| | 1536 | 1.903 | 3153 | 0.088 | | |
| | 3072 | 1.903 | 5194 | 0.165 | | |
| Reformer | 384 | 1.157 | 1681 | 0.030 | (11, 11, 7, 4, 11) | 8.8 |
| | 768 | 1.157 | 2301 | 0.046 | | |
| | 1536 | 1.157 | 5793 | 0.102 | | |
| | 3072 | 1.157 | / | / | | |
| Pyraformer | 384 | 1.308 | 2047 | 0.046 | (9, 10, 12, 8, 6) | 9.0 |
| | 768 | 1.996 | 6077 | 0.119 | | |
| | 1536 | 3.372 | / | / | | |
| | 3072 | / | / | / | | |
| LSSL | 384 | 0.121 | 1135 | 0.010 | (8, 12, 10, 7, 13) | 10.0 |
| | 768 | 0.220 | 1139 | 0.011 | | |
| | 1536 | 0.417 | 1147 | 0.013 | | |
| | 3072 | 0.812 | 1197 | 0.032 | | |
| TCN | 384 | 0.372 | 1195 | 0.020 | (12, 7, 11, 11, 12) | 10.6 |
| | 768 | 0.372 | 1333 | 0.020 | | |
| | 1536 | 0.372 | 1533 | 0.025 | | |
| | 3072 | 0.372 | 1983 | 0.061 | | |
| LSTM | 384 | 0.268 | 1201 | 0.064 | (13, 13, 13, 13, 10) | 12.4 |
| | 768 | 0.268 | 1323 | 0.122 | | |
| | 1536 | 0.268 | 1539 | 0.229 | | |
| | 3072 | 0.268 | 2017 | 0.452 | | |

## H MODEL PERFORMANCE IN MIXED DATASET

To verify the model capacity in large-scale pre-training, we evaluate the model performance when it is trained from a mixed dataset. Concretely, we mixed the hourly-collected ETTh1, ETTh2 and the 15-minute collected ETTm1, ETTm2 as the mixed dataset. Note that this mixed dataset contains diverse temporal patterns and periodicities in different data instances, making the unified training challenging. From Table 12, we can find that TimesNet can handle this mixed dataset well and generally promote the model performance in four independent subsets.

Besides, we can also find that except TimesNet, for other baselines, the mixed training may decrease the model performance in some subsets, indicating that other baselines cannot handle the complex periodicities in the mixed dataset. These results also verify the potential of TimesNet in performing as the general-purpose backbone for large-scale pre-training in time series.

Table 12: Comparison between unified training and independent training for imputation task.

| | Datasets | | ETTm1 | | | | ETTm2 | | | | ETTh1 | | | | ETTh2 | | | |
|---|---|---|---|---|---|---|---|---|---|---|---|---|---|---|---|---|---|---|
| | Mask Ratio | | 12.5% | 25% | 37.5% | 50% | 12.5% | 25% | 37.5% | 50% | 12.5% | 25% | 37.5% | 50% | 12.5% | 25% | 37.5% | 50% |
| **Autoformer** | Unified | MSE | 0.034 | 0.048 | 0.060 | 0.078 | 0.023 | 0.027 | 0.030 | 0.034 | 0.066 | 0.086 | 0.114 | 0.133 | 0.042 | 0.049 | 0.055 | 0.065 |
| | | MAE | 0.122 | 0.146 | 0.163 | 0.185 | 0.091 | 0.102 | 0.109 | 0.117 | 0.174 | 0.200 | 0.229 | 0.247 | 0.135 | 0.147 | 0.157 | 0.171 |
| | Independent | MSE | 0.034 | 0.046 | 0.057 | 0.067 | 0.023 | 0.026 | 0.030 | 0.035 | 0.074 | 0.090 | 0.109 | 0.137 | 0.044 | 0.050 | 0.060 | 0.068 |
| | | MAE | 0.124 | 0.144 | 0.161 | 0.174 | 0.092 | 0.101 | 0.108 | 0.119 | 0.182 | 0.203 | 0.222 | 0.248 | 0.138 | 0.149 | 0.163 | 0.173 |
| **FEDformer** | Unified | MSE | 0.041 | 0.057 | 0.073 | 0.099 | 0.060 | 0.089 | 0.125 | 0.172 | 0.077 | 0.101 | 0.130 | 0.164 | 0.087 | 0.125 | 0.161 | 0.214 |
| | | MAE | 0.143 | 0.169 | 0.192 | 0.224 | 0.166 | 0.205 | 0.244 | 0.287 | 0.196 | 0.228 | 0.258 | 0.289 | 0.204 | 0.246 | 0.283 | 0.326 |
| | Independent | MSE | 0.035 | 0.052 | 0.069 | 0.089 | 0.056 | 0.080 | 0.110 | 0.156 | 0.070 | 0.106 | 0.124 | 0.165 | 0.095 | 0.137 | 0.187 | 0.232 |
| | | MAE | 0.135 | 0.166 | 0.191 | 0.218 | 0.159 | 0.195 | 0.231 | 0.276 | 0.190 | 0.236 | 0.258 | 0.299 | 0.212 | 0.258 | 0.304 | 0.341 |
| **TimesNet** | Unified | MSE | **0.019** | **0.023** | **0.028** | **0.037** | **0.018** | **0.020** | **0.022** | **0.025** | **0.035** | **0.046** | **0.057** | **0.075** | **0.032** | **0.036** | **0.040** | **0.047** |
| | | MAE | **0.091** | **0.099** | **0.109** | **0.123** | **0.075** | **0.081** | **0.086** | **0.095** | **0.126** | **0.144** | **0.159** | **0.181** | **0.112** | **0.119** | **0.129** | **0.140** |
| | Independent | MSE | 0.019 | 0.023 | 0.029 | 0.037 | 0.018 | 0.020 | 0.023 | 0.026 | 0.057 | 0.069 | 0.084 | 0.102 | 0.040 | 0.046 | 0.052 | 0.060 |
| | | MAE | 0.092 | 0.101 | 0.111 | 0.124 | 0.080 | 0.085 | 0.091 | 0.098 | 0.159 | 0.178 | 0.196 | 0.215 | 0.130 | 0.141 | 0.151 | 0.162 |

## I FULL RESULTS

Due to the space limitation of the main text, we place the full results of all experiments in the following: long-term forecasting in Table 13, short-term forecasting in Table 14, imputation in Table 16, classification in Table 17 and anomaly detection in Table 15.

Table 13: Full results for the long-term forecasting task. We compare extensive competitive models under different prediction lengths. The input sequence length is set to 36 for the ILI dataset and 96 for the others. *Avg* is averaged from all four prediction lengths.

| Models | | TimesNet (Ours) | | ETSformer (2022) | | LightTS* (2022) | | DLinear* (2023) | | FEDformer (2022) | | Stationary (2022a) | | Autoformer (2021) | | Pyraformer (2021a) | | Informer (2021) | | LogTrans (2019) | | Reformer (2020) | | LSSL (2022) | | LSTM (1997) | |
|---|---|---|---|---|---|---|---|---|---|---|---|---|---|---|---|---|---|---|---|---|---|---|---|---|---|---|---|
| Metric | | MSE | MAE | MSE | MAE | MSE | MAE | MSE | MAE | MSE | MAE | MSE | MAE | MSE | MAE | MSE | MAE | MSE | MAE | MSE | MAE | MSE | MAE | MSE | MAE | MSE | MAE |
| ETTm1 | 96 | **0.338** | 0.375 | 0.375 | 0.398 | 0.374 | 0.400 | 0.345 | 0.372 | 0.379 | 0.419 | 0.386 | 0.398 | 0.505 | 0.475 | 0.543 | 0.510 | 0.672 | 0.571 | 0.600 | 0.546 | 0.538 | 0.528 | 0.450 | 0.477 | 0.863 | 0.664 |
| | 192 | **0.374** | **0.387** | 0.408 | 0.410 | 0.400 | 0.407 | 0.380 | 0.389 | 0.426 | 0.441 | 0.459 | 0.444 | 0.553 | 0.496 | 0.557 | 0.537 | 0.795 | 0.669 | 0.837 | 0.700 | 0.658 | 0.592 | 0.469 | 0.481 | 1.113 | 0.776 |
| | 336 | **0.410** | **0.411** | 0.435 | 0.428 | 0.438 | 0.438 | 0.413 | 0.413 | 0.445 | 0.459 | 0.495 | 0.464 | 0.621 | 0.537 | 0.754 | 0.655 | 1.212 | 0.871 | 1.124 | 0.832 | 0.898 | 0.721 | 0.583 | 0.574 | 1.267 | 0.832 |
| | 720 | 0.478 | **0.450** | 0.499 | 0.462 | 0.527 | 0.502 | 0.474 | 0.453 | 0.543 | 0.490 | 0.585 | 0.516 | 0.671 | 0.561 | 0.908 | 0.724 | 1.166 | 0.823 | 1.153 | 0.820 | 1.102 | 0.841 | 0.632 | 0.596 | 1.324 | 0.858 |
| | Avg | **0.400** | **0.406** | 0.429 | 0.425 | 0.435 | 0.437 | 0.403 | 0.407 | 0.448 | 0.452 | 0.481 | 0.456 | 0.588 | 0.517 | 0.691 | 0.607 | 0.961 | 0.734 | 0.929 | 0.725 | 0.799 | 0.671 | 0.533 | 0.532 | 1.142 | 0.782 |
| ETTm2 | 96 | **0.187** | **0.267** | 0.189 | 0.280 | 0.209 | 0.308 | 0.193 | 0.292 | 0.203 | 0.287 | 0.192 | 0.274 | 0.255 | 0.339 | 0.435 | 0.507 | 0.365 | 0.453 | 0.768 | 0.642 | 0.658 | 0.619 | 0.243 | 0.342 | 2.041 | 1.073 |
| | 192 | **0.249** | **0.309** | 0.253 | 0.319 | 0.311 | 0.382 | 0.284 | 0.362 | 0.269 | 0.328 | 0.280 | 0.339 | 0.281 | 0.340 | 0.730 | 0.673 | 0.533 | 0.563 | 0.989 | 0.757 | 1.078 | 0.827 | 0.392 | 0.448 | 2.249 | 1.112 |
| | 336 | 0.321 | 0.351 | 0.314 | 0.357 | 0.442 | 0.466 | 0.369 | 0.427 | 0.325 | 0.366 | 0.334 | 0.361 | 0.339 | 0.372 | 1.201 | 0.845 | 1.363 | 0.887 | 1.334 | 0.872 | 1.549 | 0.972 | 0.932 | 0.724 | 2.568 | 1.238 |
| | 720 | **0.408** | **0.403** | 0.414 | 0.413 | 0.675 | 0.587 | 0.554 | 0.522 | 0.421 | 0.415 | 0.417 | 0.413 | 0.433 | 0.432 | 3.625 | 1.451 | 3.379 | 1.338 | 3.048 | 1.328 | 2.631 | 1.242 | 1.372 | 0.879 | 2.720 | 1.287 |
| | Avg | **0.291** | **0.333** | 0.293 | 0.342 | 0.409 | 0.436 | 0.350 | 0.401 | 0.305 | 0.349 | 0.306 | 0.347 | 0.327 | 0.371 | 1.498 | 0.869 | 1.410 | 0.810 | 1.535 | 0.900 | 1.479 | 0.915 | 0.735 | 0.598 | 2.395 | 1.177 |
| ETTh1 | 96 | 0.384 | 0.402 | 0.494 | 0.479 | 0.424 | 0.432 | 0.386 | **0.400** | **0.376** | 0.419 | 0.513 | 0.491 | 0.449 | 0.459 | 0.664 | 0.612 | 0.865 | 0.713 | 0.878 | 0.740 | 0.837 | 0.728 | 0.548 | 0.528 | 1.044 | 0.773 |
| | 192 | 0.436 | 0.429 | 0.538 | 0.504 | 0.475 | 0.462 | 0.437 | 0.432 | **0.420** | 0.448 | 0.534 | 0.504 | 0.500 | 0.482 | 0.790 | 0.681 | 1.008 | 0.792 | 1.037 | 0.824 | 0.923 | 0.766 | 0.542 | 0.526 | 1.217 | 0.832 |
| | 336 | 0.491 | 0.469 | 0.574 | 0.521 | 0.518 | 0.488 | 0.481 | **0.459** | 0.459 | 0.465 | 0.588 | 0.535 | 0.521 | 0.496 | 0.891 | 0.738 | 1.107 | 0.809 | 1.238 | 0.932 | 1.097 | 0.835 | 1.298 | 0.942 | 1.259 | 0.841 |
| | 720 | 0.521 | **0.500** | 0.562 | 0.535 | 0.547 | 0.533 | 0.519 | 0.516 | **0.506** | 0.507 | 0.643 | 0.616 | 0.514 | 0.512 | 0.963 | 0.782 | 1.181 | 0.865 | 1.135 | 0.852 | 1.257 | 0.889 | 0.721 | 0.659 | 1.271 | 0.838 |
| | Avg | 0.458 | **0.450** | 0.542 | 0.510 | 0.491 | 0.479 | 0.456 | 0.452 | **0.440** | 0.460 | 0.570 | 0.537 | 0.496 | 0.487 | 0.827 | 0.703 | 1.040 | 0.795 | 1.072 | 0.837 | 1.029 | 0.805 | 0.777 | 0.664 | 1.198 | 0.821 |
| ETTh2 | 96 | 0.340 | 0.374 | 0.340 | 0.391 | 0.397 | 0.437 | **0.333** | **0.387** | 0.358 | 0.397 | 0.476 | 0.458 | 0.346 | 0.388 | 0.645 | 0.597 | 3.755 | 1.525 | 2.116 | 1.197 | 2.626 | 1.317 | 1.616 | 1.036 | 2.522 | 1.278 |
| | 192 | **0.402** | **0.414** | 0.430 | 0.439 | 0.520 | 0.504 | 0.477 | 0.476 | 0.429 | 0.439 | 0.512 | 0.493 | 0.456 | 0.452 | 0.788 | 0.683 | 5.602 | 1.931 | 4.315 | 1.635 | 11.12 | 2.979 | 2.083 | 1.197 | 3.312 | 1.384 |
| | 336 | **0.452** | **0.452** | 0.485 | 0.479 | 0.626 | 0.559 | 0.594 | 0.541 | 0.496 | 0.487 | 0.552 | 0.551 | 0.482 | 0.486 | 0.907 | 0.747 | 4.721 | 1.835 | 1.124 | 1.604 | 9.323 | 2.769 | 2.970 | 1.439 | 3.291 | 1.388 |
| | 720 | **0.462** | **0.468** | 0.500 | 0.497 | 0.863 | 0.672 | 0.831 | 0.657 | 0.463 | 0.474 | 0.562 | 0.560 | 0.515 | 0.511 | 0.963 | 0.783 | 3.647 | 1.625 | 3.188 | 1.540 | 3.874 | 1.697 | 2.576 | 1.363 | 3.257 | 1.357 |
| | Avg | **0.414** | **0.427** | 0.439 | 0.452 | 0.602 | 0.543 | 0.559 | 0.515 | 0.437 | 0.449 | 0.526 | 0.516 | 0.450 | 0.459 | 0.826 | 0.703 | 4.431 | 1.729 | 2.686 | 1.494 | 6.736 | 2.191 | 2.311 | 1.259 | 3.095 | 1.352 |
| Electricity | 96 | **0.168** | **0.272** | 0.187 | 0.304 | 0.207 | 0.307 | 0.197 | 0.282 | 0.193 | 0.308 | 0.169 | 0.273 | 0.201 | 0.317 | 0.386 | 0.449 | 0.274 | 0.368 | 0.258 | 0.357 | 0.312 | 0.402 | 0.300 | 0.392 | 0.375 | 0.437 |
| | 192 | 0.184 | 0.289 | 0.199 | 0.315 | 0.213 | 0.316 | 0.196 | **0.285** | 0.201 | 0.315 | 0.182 | 0.286 | 0.222 | 0.334 | 0.378 | 0.443 | 0.296 | 0.386 | 0.266 | 0.368 | 0.348 | 0.433 | 0.297 | 0.390 | 0.442 | 0.473 |
| | 336 | **0.198** | **0.300** | 0.212 | 0.329 | 0.230 | 0.333 | 0.209 | 0.301 | 0.214 | 0.329 | 0.200 | 0.304 | 0.231 | 0.338 | 0.376 | 0.443 | 0.300 | 0.394 | 0.280 | 0.380 | 0.350 | 0.433 | 0.317 | 0.403 | 0.439 | 0.473 |
| | 720 | **0.220** | **0.320** | 0.233 | 0.345 | 0.265 | 0.360 | 0.245 | 0.333 | 0.246 | 0.355 | 0.222 | 0.321 | 0.254 | 0.361 | 0.376 | 0.445 | 0.373 | 0.439 | 0.283 | 0.376 | 0.340 | 0.420 | 0.338 | 0.417 | 0.980 | 0.814 |
| | Avg | **0.192** | **0.295** | 0.208 | 0.323 | 0.229 | 0.329 | 0.212 | 0.300 | 0.214 | 0.327 | 0.193 | 0.296 | 0.227 | 0.338 | 0.379 | 0.445 | 0.311 | 0.397 | 0.272 | 0.370 | 0.338 | 0.422 | 0.313 | 0.401 | 0.559 | 0.549 |
| Traffic | 96 | 0.593 | 0.321 | 0.607 | 0.392 | 0.615 | 0.391 | 0.650 | 0.396 | **0.587** | 0.366 | 0.612 | 0.338 | 0.613 | 0.388 | 0.867 | 0.468 | 0.719 | 0.391 | 0.684 | 0.384 | 0.732 | 0.423 | 0.798 | 0.436 | 0.843 | 0.453 |
| | 192 | 0.617 | **0.336** | 0.621 | 0.399 | 0.601 | 0.382 | 0.598 | 0.370 | 0.604 | 0.373 | 0.613 | 0.340 | 0.616 | 0.382 | 0.869 | 0.467 | 0.696 | 0.379 | 0.685 | 0.390 | 0.733 | 0.420 | 0.849 | 0.481 | 0.847 | 0.453 |
| | 336 | 0.629 | 0.336 | 0.622 | 0.396 | 0.613 | 0.386 | 0.605 | 0.373 | 0.621 | 0.383 | 0.618 | **0.328** | 0.622 | 0.337 | 0.881 | 0.469 | 0.777 | 0.420 | 0.734 | 0.408 | 0.742 | 0.420 | 0.828 | 0.476 | 0.853 | 0.455 |
| | 720 | 0.640 | **0.350** | 0.632 | 0.396 | 0.658 | 0.407 | 0.645 | 0.394 | 0.626 | 0.382 | 0.653 | 0.355 | 0.660 | 0.408 | 0.896 | 0.473 | 0.864 | 0.472 | 0.717 | 0.396 | 0.755 | 0.423 | 0.854 | 0.489 | 1.500 | 0.805 |
| | Avg | 0.620 | **0.336** | 0.621 | 0.396 | 0.622 | 0.392 | 0.625 | 0.383 | **0.610** | 0.376 | 0.624 | 0.340 | 0.628 | 0.379 | 0.878 | 0.469 | 0.764 | 0.416 | 0.705 | 0.395 | 0.741 | 0.422 | 0.832 | 0.471 | 1.011 | 0.541 |
| Weather | 96 | **0.172** | **0.220** | 0.197 | 0.281 | 0.182 | 0.242 | 0.196 | 0.255 | 0.217 | 0.296 | 0.173 | 0.223 | 0.266 | 0.336 | 0.622 | 0.556 | 0.300 | 0.384 | 0.458 | 0.490 | 0.689 | 0.596 | 0.174 | 0.252 | 0.369 | 0.406 |
| | 192 | **0.219** | **0.261** | 0.237 | 0.312 | 0.227 | 0.287 | 0.237 | 0.296 | 0.276 | 0.336 | 0.245 | 0.285 | 0.307 | 0.367 | 0.739 | 0.624 | 0.598 | 0.544 | 0.658 | 0.589 | 0.752 | 0.638 | 0.238 | 0.313 | 0.416 | 0.435 |
| | 336 | **0.280** | **0.306** | 0.298 | 0.353 | 0.282 | 0.334 | 0.283 | 0.335 | 0.339 | 0.380 | 0.321 | 0.338 | 0.359 | 0.395 | 1.004 | 0.753 | 0.578 | 0.523 | 0.797 | 0.652 | 0.639 | 0.596 | 0.287 | 0.355 | 0.455 | 0.454 |
| | 720 | 0.365 | **0.359** | 0.352 | 0.288 | 0.352 | 0.386 | 0.345 | 0.381 | 0.403 | 0.428 | 0.414 | 0.410 | 0.419 | 0.428 | 1.420 | 0.934 | 1.059 | 0.741 | 0.869 | 0.675 | 1.130 | 0.792 | 0.384 | 0.415 | 0.535 | 0.520 |
| | Avg | **0.259** | **0.287** | 0.271 | 0.334 | 0.261 | 0.312 | 0.265 | 0.317 | 0.309 | 0.360 | 0.288 | 0.314 | 0.338 | 0.382 | 0.946 | 0.717 | 0.634 | 0.548 | 0.696 | 0.602 | 0.803 | 0.656 | 0.271 | 0.334 | 0.444 | 0.454 |
| Exchange | 96 | 0.107 | 0.234 | **0.085** | **0.204** | 0.116 | 0.262 | 0.088 | 0.218 | 0.148 | 0.278 | 0.111 | 0.237 | 0.197 | 0.323 | 1.748 | 1.105 | 0.847 | 0.752 | 0.968 | 0.812 | 1.065 | 0.829 | 0.395 | 0.474 | 1.453 | 1.049 |
| | 192 | 0.226 | 0.344 | 0.182 | 0.303 | 0.215 | 0.359 | 0.176 | 0.315 | 0.271 | 0.380 | 0.219 | 0.335 | 0.300 | 0.369 | 1.874 | 1.151 | 1.204 | 0.895 | 1.040 | 0.851 | 1.188 | 0.906 | 0.776 | 0.698 | 1.846 | 1.179 |
| | 336 | 0.367 | 0.448 | 0.348 | 0.428 | 0.377 | 0.466 | 0.313 | 0.427 | 0.460 | 0.500 | 0.421 | 0.476 | 0.509 | 0.524 | 1.943 | 1.172 | 1.672 | 1.036 | 1.659 | 1.081 | 1.357 | 0.976 | 1.029 | 0.797 | 2.136 | 1.231 |
| | 720 | 0.964 | 0.746 | 1.025 | 0.774 | 0.831 | 0.699 | 0.839 | 0.695 | 1.195 | 0.841 | 1.092 | 0.769 | 1.447 | 0.941 | 2.085 | 1.206 | 2.478 | 1.310 | 1.941 | 1.127 | 1.510 | 1.016 | 2.283 | 1.222 | 2.984 | 1.427 |
| | Avg | 0.416 | 0.443 | 0.410 | 0.427 | 0.385 | 0.447 | 0.354 | 0.414 | 0.519 | 0.500 | 0.461 | 0.454 | 0.613 | 0.539 | 1.913 | 1.159 | 1.550 | 0.998 | 1.402 | 0.968 | 1.280 | 0.932 | 1.121 | 0.798 | 2.105 | 1.221 |
| ILI | 24 | 2.317 | **0.934** | 2.527 | 1.020 | 8.313 | 2.144 | 2.398 | 1.040 | 3.228 | 1.260 | 2.294 | 0.945 | 3.483 | 1.287 | 7.394 | 2.012 | 5.764 | 1.677 | 4.480 | 1.444 | 4.400 | 1.382 | 4.381 | 1.425 | 5.914 | 1.734 |
| | 36 | 1.972 | 0.920 | 2.615 | 1.007 | 6.631 | 1.902 | 2.646 | 1.088 | 2.679 | 1.080 | **1.825** | **0.848** | 3.103 | 1.148 | 7.551 | 2.031 | 4.755 | 1.467 | 4.799 | 1.467 | 4.783 | 1.448 | 4.442 | 1.416 | 6.631 | 1.845 |
| | 48 | 2.238 | 0.940 | 2.359 | 0.972 | 7.299 | 1.982 | 2.614 | 1.086 | 2.622 | 1.078 | **2.010** | **0.900** | 2.669 | 1.085 | 7.662 | 2.057 | 4.763 | 1.469 | 4.800 | 1.468 | 4.832 | 1.465 | 4.559 | 1.443 | 6.736 | 1.857 |
| | 60 | **2.027** | **0.928** | 2.487 | 1.016 | 7.283 | 1.985 | 2.804 | 1.146 | 2.857 | 1.157 | 2.178 | 0.963 | 2.770 | 1.125 | 7.931 | 2.100 | 5.264 | 1.564 | 5.278 | 1.560 | 4.882 | 1.483 | 4.651 | 1.474 | 6.870 | 1.879 |
| | Avg | 2.139 | 0.931 | 2.497 | 1.004 | 7.382 | 2.003 | 2.616 | 1.090 | 2.847 | 1.144 | **2.077** | **0.914** | 3.006 | 1.161 | 7.635 | 2.050 | 5.137 | 1.544 | 4.839 | 1.485 | 4.724 | 1.445 | 4.508 | 1.440 | 6.538 | 1.829 |
| 1st Count | | **40** | | 4 | | 1 | | 14 | | 6 | | 7 | | 0 | | 0 | | 0 | | 0 | | 0 | | 0 | | 0 | |

∗ means that there are some mismatches between our input-output setting and their papers. We adopt their official codes and only change the length of input and output sequences for a fair comparison.

Table 14: Full results for the short-term forecasting task in the M4 dataset. ∗. in the Transformers indicates the name of ∗former. *Stationary* means the Non-stationary Transformer.

| Models | | TimesNet (Ours) | N-HiTS (2022) | N-BEATS* (2019) | ETS. (2022) | LightTS (2022) | DLinear (2023) | FED. (2022) | Stationary (2022a) | Auto. (2021) | Pyra. (2021a) | In. (2021) | LogTrans (2019) | Re. (2020) | LSTM (1997) | TCN (2019) | LSSL (2022) |
|---|---|---|---|---|---|---|---|---|---|---|---|---|---|---|---|---|---|
| Yearly | SMAPE | **13.387** | 13.418 | 13.436 | 18.009 | 14.247 | 16.965 | 13.728 | 13.717 | 13.974 | 15.530 | 14.727 | 17.107 | 16.169 | 176.040 | 14.920 | 61.675 |
| | MASE | **2.996** | 3.045 | 3.043 | 4.487 | 3.109 | 4.283 | 3.048 | 3.078 | 3.134 | 3.711 | 3.418 | 4.177 | 3.800 | 31.033 | 3.364 | 19.953 |
| | OWA | **0.786** | 0.793 | 0.794 | 1.115 | 0.827 | 1.058 | 0.803 | 0.807 | 0.822 | 0.942 | 0.881 | 1.049 | 0.973 | 9.290 | 0.880 | 4.397 |
| Quarterly | SMAPE | **10.100** | 10.202 | 10.124 | 13.376 | 11.364 | 12.145 | 10.792 | 10.958 | 11.338 | 15.449 | 11.360 | 13.207 | 13.313 | 172.808 | 11.122 | 65.999 |
| | MASE | 1.182 | 1.194 | **1.169** | 1.906 | 1.328 | 1.520 | 1.283 | 1.325 | 1.365 | 2.350 | 1.401 | 1.827 | 1.775 | 19.753 | 1.360 | 17.662 |
| | OWA | 0.890 | 0.899 | **0.886** | 1.302 | 1.000 | 1.106 | 0.958 | 0.981 | 1.012 | 1.558 | 1.027 | 1.266 | 1.252 | 15.049 | 1.001 | 9.436 |
| Monthly | SMAPE | **12.670** | 12.791 | 12.677 | 14.588 | 14.014 | 13.514 | 14.260 | 13.917 | 13.958 | 17.642 | 14.062 | 16.149 | 20.128 | 143.237 | 15.626 | 64.664 |
| | MASE | **0.933** | 0.969 | 0.937 | 1.368 | 1.053 | 1.037 | 1.102 | 1.097 | 1.103 | 1.913 | 1.141 | 1.660 | 2.614 | 16.551 | 1.274 | 16.245 |
| | OWA | **0.878** | 0.899 | 0.880 | 1.149 | 0.981 | 0.956 | 1.012 | 0.998 | 1.002 | 1.511 | 1.024 | 1.340 | 1.927 | 12.747 | 1.141 | 9.879 |
| Others | SMAPE | **4.891** | 5.061 | 4.925 | 7.267 | 15.880 | 6.709 | 4.954 | 6.302 | 5.485 | 24.786 | 24.460 | 23.236 | 32.491 | 186.282 | 7.186 | 121.844 |
| | MASE | 3.302 | **3.216** | 3.391 | 5.240 | 11.434 | 4.953 | 3.264 | 4.064 | 3.865 | 18.581 | 20.960 | 16.288 | 33.355 | 119.294 | 4.677 | 91.650 |
| | OWA | **1.035** | 1.040 | 1.053 | 1.591 | 3.474 | 1.487 | 1.036 | 1.304 | 1.187 | 5.538 | 5.879 | 5.013 | 8.679 | 38.411 | 1.494 | 27.273 |
| Weighted Average | SMAPE | **11.829** | 11.927 | 11.851 | 14.718 | 13.525 | 13.639 | 12.840 | 12.780 | 12.909 | 16.987 | 14.086 | 16.018 | 18.200 | 160.031 | 13.961 | 67.156 |
| | MASE | **1.585** | 1.613 | 1.599 | 2.408 | 2.111 | 2.095 | 1.701 | 1.756 | 1.771 | 3.265 | 2.718 | 3.010 | 4.223 | 25.788 | 1.945 | 21.208 |
| | OWA | **0.851** | 0.861 | 0.855 | 1.172 | 1.051 | 1.051 | 0.918 | 0.930 | 0.939 | 1.480 | 1.230 | 1.378 | 1.775 | 12.642 | 1.023 | 8.021 |

∗ The original paper of N-BEATS (Oreshkin et al., 2019) adopts a special ensemble method to promote the performance. For fair comparisons, we remove the ensemble and only compare the pure forecasting models.

Table 15: Full results for the anomaly detection task. The P, R and F1 represent the precision, recall and F1-score (%) respectively. F1-score is the harmonic mean of precision and recall. A higher value of P, R and F1 indicates a better performance.

| Datasets | | SMD | | | MSL | | | SMAP | | | SWaT | | | PSM | | | Avg F1 |
|---|---|---|---|---|---|---|---|---|---|---|---|---|---|---|---|---|---|
| Metrics | | P | R | F1 | P | R | F1 | P | R | F1 | P | R | F1 | P | R | F1 | (%) |
| LSTM | (1997) | 78.52 | 65.47 | 71.41 | 78.04 | 86.22 | 81.93 | 91.06 | 57.49 | 70.48 | 78.06 | 91.72 | 84.34 | 69.24 | 99.53 | 81.67 | 77.97 |
| Transformer | (2017) | 83.58 | 76.13 | 79.56 | 71.57 | 87.37 | 78.68 | 89.37 | 57.12 | 69.70 | 68.84 | 96.53 | 80.37 | 62.75 | 96.56 | 76.07 | 76.88 |
| LogTrans | (2019) | 83.46 | 70.13 | 76.21 | 73.05 | 87.37 | 79.57 | 89.15 | 57.59 | 69.97 | 68.67 | 97.32 | 80.52 | 63.06 | 98.00 | 76.74 | 76.60 |
| TCN | (2019) | 84.06 | 79.07 | 81.49 | 75.11 | 82.44 | 78.60 | 86.90 | 59.23 | 70.45 | 76.59 | 95.71 | 85.09 | 54.59 | 99.77 | 70.57 | 77.24 |
| Reformer | (2020) | 82.58 | 69.24 | 75.32 | 85.51 | 83.31 | 84.40 | 90.91 | 57.44 | 70.40 | 72.50 | 96.53 | 82.80 | 59.93 | 95.38 | 73.61 | 77.31 |
| Informer | (2021) | 86.60 | 77.23 | 81.65 | 81.77 | 86.48 | 84.06 | 90.11 | 57.13 | 69.92 | 70.29 | 96.75 | 81.43 | 64.27 | 96.33 | 77.10 | 78.83 |
| Anomaly* | (2021) | 88.91 | 82.23 | 85.49 | 79.61 | 87.37 | 83.31 | 91.85 | 58.11 | 71.18 | 72.51 | 97.32 | 83.10 | 68.35 | 94.72 | 79.40 | 80.50 |
| Pyraformer | (2021a) | 85.61 | 80.61 | 83.04 | 83.81 | 85.93 | 84.86 | 92.54 | 57.71 | 71.09 | 87.92 | 96.00 | 91.78 | 71.67 | 96.02 | 82.08 | 82.57 |
| Autoformer | (2021) | 88.06 | 82.35 | 85.11 | 77.27 | 80.92 | 79.05 | 90.40 | 58.62 | 71.12 | 89.85 | 95.81 | 92.74 | 99.08 | 88.15 | 93.29 | 84.26 |
| LSSL | (2022) | 78.51 | 65.32 | 71.31 | 77.55 | 88.18 | 82.53 | 89.43 | 53.43 | 66.90 | 79.05 | 93.72 | 85.76 | 66.02 | 92.93 | 77.20 | 76.74 |
| Stationary | (2022a) | 88.33 | 81.21 | 84.62 | 68.55 | 89.14 | 77.50 | 89.37 | 59.02 | 71.09 | 68.03 | 96.75 | 79.88 | 97.82 | 96.76 | 97.29 | 82.08 |
| DLinear | (2023) | 83.62 | 71.52 | 77.10 | 84.34 | 85.42 | 84.88 | 92.32 | 55.41 | 69.26 | 80.91 | 95.30 | 87.52 | 98.28 | 89.26 | 93.55 | 82.46 |
| ETSformer | (2022) | 87.44 | 79.23 | 83.13 | 85.13 | 84.93 | 85.03 | 92.25 | 55.75 | 69.50 | 90.02 | 80.36 | 84.91 | 99.31 | 85.28 | 91.76 | 82.87 |
| LightTS | (2022) | 87.10 | 78.42 | 82.53 | 82.40 | 75.78 | 78.95 | 92.58 | 55.27 | 69.21 | 91.98 | 94.72 | 93.33 | 98.37 | 95.97 | 97.15 | 84.23 |
| FEDformer | (2022) | 87.95 | 82.39 | 85.08 | 77.14 | 80.07 | 78.57 | 90.47 | 58.10 | 70.76 | 90.17 | 96.42 | 93.19 | 97.31 | 97.16 | 97.23 | 84.97 |
| TimesNet | **(Inception)** | 87.76 | 82.63 | 85.12 | 82.97 | 85.42 | 84.18 | 91.50 | 57.80 | 70.85 | 88.31 | 96.24 | 92.10 | 98.22 | 92.21 | 95.21 | 85.49 |
| TimesNet | **(ResNeXt)** | 88.66 | 83.14 | **85.81** | 83.92 | 86.42 | **85.15** | 92.52 | 58.29 | **71.52** | 86.76 | 97.32 | 91.74 | 98.19 | 96.76 | **97.47** | **86.34** |

∗ The original paper of Anomaly Transformer (Xu et al., 2021) adopts the temporal association and reconstruction error as a joint anomaly criterion. For fair comparisons, we only use reconstruction error here.

Table 16: Full results for the imputation task. We randomly mask 12.5%, 25%, 37.5% and 50% time points to compare the model performance under different missing degrees. ∗. in the Transformers indicates the name of ∗former.

| Models | TimesNet (Ours) | | ETS. (2022) | | LightTS* (2022) | | DLinear* (2023) | | FED. (2022) | | Stationary (2022a) | | Auto. (2021) | | Pyra. (2021a) | | In. (2021) | | LogTrans (2019) | | Re. (2020) | | LSTM (1997) | | TCN (2019) | | LSSL (2022) | |
|---|---|---|---|---|---|---|---|---|---|---|---|---|---|---|---|---|---|---|---|---|---|---|---|---|---|---|---|---|
| Mask Ratio | MSE | MAE | MSE | MAE | MSE | MAE | MSE | MAE | MSE | MAE | MSE | MAE | MSE | MAE | MSE | MAE | MSE | MAE | MSE | MAE | MSE | MAE | MSE | MAE | MSE | MAE | MSE | MAE |
| **ETTm1** 12.5% | **0.019** | **0.092** | 0.067 | 0.188 | 0.075 | 0.180 | 0.058 | 0.162 | 0.035 | 0.135 | 0.026 | 0.107 | 0.034 | 0.124 | 0.670 | 0.541 | 0.047 | 0.155 | 0.041 | 0.141 | 0.032 | 0.126 | 0.974 | 0.780 | 0.510 | 0.493 | 0.101 | 0.231 |
| 25% | **0.023** | **0.101** | 0.096 | 0.229 | 0.093 | 0.206 | 0.080 | 0.193 | 0.052 | 0.166 | 0.032 | 0.119 | 0.046 | 0.144 | 0.689 | 0.553 | 0.063 | 0.180 | 0.044 | 0.144 | 0.042 | 0.146 | 1.032 | 0.807 | 0.518 | 0.500 | 0.106 | 0.235 |
| 37.5% | **0.029** | **0.111** | 0.133 | 0.271 | 0.113 | 0.231 | 0.103 | 0.219 | 0.069 | 0.191 | 0.039 | 0.131 | 0.057 | 0.161 | 0.737 | 0.581 | 0.079 | 0.200 | 0.052 | 0.158 | 0.063 | 0.182 | 0.999 | 0.792 | 0.516 | 0.499 | 0.116 | 0.246 |
| 50% | **0.036** | **0.124** | 0.186 | 0.323 | 0.134 | 0.255 | 0.132 | 0.248 | 0.089 | 0.218 | 0.047 | 0.145 | 0.067 | 0.174 | 0.770 | 0.605 | 0.093 | 0.218 | 0.063 | 0.173 | 0.082 | 0.208 | 0.952 | 0.763 | 0.519 | 0.496 | 0.129 | 0.260 |
| Avg | **0.027** | **0.107** | 0.120 | 0.253 | 0.104 | 0.218 | 0.093 | 0.206 | 0.062 | 0.177 | 0.036 | 0.126 | 0.051 | 0.150 | 0.717 | 0.570 | 0.071 | 0.188 | 0.050 | 0.154 | 0.055 | 0.166 | 0.989 | 0.786 | 0.516 | 0.497 | 0.113 | 0.254 |
| **ETTm2** 12.5% | **0.018** | **0.080** | 0.108 | 0.239 | 0.034 | 0.127 | 0.062 | 0.166 | 0.056 | 0.159 | 0.021 | 0.088 | 0.023 | 0.092 | 0.394 | 0.470 | 0.133 | 0.270 | 0.103 | 0.229 | 0.108 | 0.228 | 1.013 | 0.805 | 0.307 | 0.441 | 0.150 | 0.298 |
| 25% | **0.020** | **0.085** | 0.164 | 0.294 | 0.042 | 0.143 | 0.085 | 0.196 | 0.080 | 0.195 | 0.024 | 0.096 | 0.026 | 0.101 | 0.421 | 0.482 | 0.135 | 0.272 | 0.120 | 0.248 | 0.136 | 0.262 | 1.039 | 0.814 | 0.263 | 0.402 | 0.159 | 0.306 |
| 37.5% | **0.023** | **0.091** | 0.237 | 0.356 | 0.051 | 0.159 | 0.106 | 0.222 | 0.110 | 0.231 | 0.027 | 0.103 | 0.030 | 0.108 | 0.478 | 0.521 | 0.155 | 0.293 | 0.138 | 0.260 | 0.175 | 0.300 | 0.917 | 0.744 | 0.250 | 0.396 | 0.180 | 0.321 |
| 50% | **0.026** | **0.098** | 0.323 | 0.421 | 0.059 | 0.174 | 0.131 | 0.247 | 0.156 | 0.276 | 0.030 | 0.108 | 0.035 | 0.119 | 0.568 | 0.560 | 0.200 | 0.333 | 0.117 | 0.247 | 0.211 | 0.329 | 1.140 | 0.835 | 0.246 | 0.389 | 0.210 | 0.353 |
| Avg | **0.022** | **0.088** | 0.208 | 0.327 | 0.046 | 0.151 | 0.096 | 0.208 | 0.101 | 0.215 | 0.026 | 0.099 | 0.029 | 0.105 | 0.465 | 0.508 | 0.156 | 0.292 | 0.119 | 0.246 | 0.157 | 0.280 | 1.027 | 0.800 | 0.266 | 0.407 | 0.175 | 0.324 |
| **ETTh1** 12.5% | **0.057** | **0.159** | 0.126 | 0.263 | 0.240 | 0.345 | 0.151 | 0.267 | 0.070 | 0.190 | 0.060 | 0.165 | 0.074 | 0.182 | 0.857 | 0.609 | 0.114 | 0.234 | 0.229 | 0.330 | 0.074 | 0.194 | 1.265 | 0.896 | 0.599 | 0.554 | 0.422 | 0.461 |
| 25% | **0.069** | **0.178** | 0.169 | 0.304 | 0.265 | 0.364 | 0.180 | 0.292 | 0.106 | 0.236 | 0.080 | 0.189 | 0.090 | 0.203 | 0.829 | 0.672 | 0.140 | 0.262 | 0.207 | 0.323 | 0.102 | 0.227 | 1.262 | 0.883 | 0.610 | 0.567 | 0.412 | 0.456 |
| 37.5% | **0.084** | **0.196** | 0.220 | 0.347 | 0.296 | 0.382 | 0.215 | 0.318 | 0.124 | 0.258 | 0.102 | 0.212 | 0.109 | 0.222 | 0.830 | 0.675 | 0.174 | 0.293 | 0.210 | 0.328 | 0.135 | 0.261 | 1.200 | 0.867 | 0.628 | 0.577 | 0.421 | 0.461 |
| 50% | **0.102** | **0.215** | 0.293 | 0.402 | 0.334 | 0.404 | 0.257 | 0.347 | 0.165 | 0.299 | 0.133 | 0.240 | 0.137 | 0.248 | 0.854 | 0.691 | 0.215 | 0.325 | 0.230 | 0.348 | 0.179 | 0.298 | 1.174 | 0.849 | 0.648 | 0.587 | 0.443 | 0.473 |
| Avg | **0.078** | **0.187** | 0.202 | 0.329 | 0.284 | 0.373 | 0.201 | 0.306 | 0.117 | 0.246 | 0.094 | 0.201 | 0.103 | 0.214 | 0.842 | 0.682 | 0.161 | 0.279 | 0.219 | 0.332 | 0.122 | 0.245 | 1.225 | 0.873 | 0.621 | 0.571 | 0.424 | 0.481 |
| **ETTh2** 12.5% | **0.040** | **0.130** | 0.187 | 0.319 | 0.101 | 0.231 | 0.100 | 0.216 | 0.095 | 0.212 | 0.042 | 0.133 | 0.044 | 0.138 | 0.976 | 0.754 | 0.305 | 0.431 | 0.173 | 0.308 | 0.163 | 0.289 | 2.060 | 1.120 | 0.410 | 0.494 | 0.521 | 0.555 |
| 25% | **0.046** | **0.141** | 0.279 | 0.390 | 0.115 | 0.246 | 0.127 | 0.247 | 0.137 | 0.258 | 0.049 | 0.147 | 0.050 | 0.149 | 1.037 | 0.774 | 0.322 | 0.444 | 0.175 | 0.310 | 0.206 | 0.331 | 2.007 | 1.105 | 0.419 | 0.490 | 0.487 | 0.535 |
| 37.5% | **0.052** | **0.151** | 0.400 | 0.465 | 0.126 | 0.257 | 0.158 | 0.276 | 0.187 | 0.304 | 0.056 | 0.158 | 0.060 | 0.163 | 1.107 | 0.800 | 0.353 | 0.462 | 0.185 | 0.315 | 0.252 | 0.370 | 2.033 | 1.111 | 0.429 | 0.498 | 0.487 | 0.529 |
| 50% | **0.060** | **0.162** | 0.602 | 0.572 | 0.136 | 0.268 | 0.183 | 0.299 | 0.232 | 0.341 | 0.065 | 0.170 | 0.068 | 0.173 | 1.193 | 0.838 | 0.369 | 0.472 | 0.212 | 0.339 | 0.316 | 0.419 | 2.054 | 1.119 | 0.467 | 0.529 | 0.484 | 0.523 |
| Avg | **0.049** | **0.146** | 0.367 | 0.436 | 0.119 | 0.250 | 0.142 | 0.259 | 0.163 | 0.279 | 0.053 | 0.152 | 0.055 | 0.156 | 1.079 | 0.792 | 0.337 | 0.452 | 0.186 | 0.318 | 0.234 | 0.352 | 2.039 | 1.114 | 0.431 | 0.503 | 0.495 | 0.475 |
| **Electricity** 12.5% | **0.085** | **0.202** | 0.196 | 0.321 | 0.102 | 0.229 | 0.092 | 0.214 | 0.107 | 0.237 | 0.093 | 0.210 | 0.089 | 0.210 | 0.297 | 0.383 | 0.218 | 0.326 | 0.164 | 0.296 | 0.190 | 0.308 | 0.277 | 0.366 | 0.621 | 0.620 | 0.217 | 0.341 |
| 25% | **0.089** | **0.206** | 0.207 | 0.332 | 0.121 | 0.252 | 0.118 | 0.247 | 0.120 | 0.251 | 0.097 | 0.214 | 0.096 | 0.220 | 0.294 | 0.380 | 0.219 | 0.326 | 0.169 | 0.299 | 0.197 | 0.312 | 0.281 | 0.369 | 0.559 | 0.585 | 0.219 | 0.341 |
| 37.5% | **0.094** | **0.213** | 0.219 | 0.344 | 0.141 | 0.273 | 0.144 | 0.276 | 0.136 | 0.266 | 0.102 | 0.220 | 0.104 | 0.229 | 0.296 | 0.381 | 0.222 | 0.328 | 0.178 | 0.305 | 0.203 | 0.315 | 0.275 | 0.364 | 0.567 | 0.588 | 0.223 | 0.343 |
| 50% | **0.100** | **0.221** | 0.235 | 0.357 | 0.160 | 0.293 | 0.175 | 0.305 | 0.158 | 0.284 | 0.108 | 0.228 | 0.113 | 0.239 | 0.299 | 0.383 | 0.228 | 0.331 | 0.187 | 0.312 | 0.210 | 0.319 | 0.273 | 0.361 | 0.581 | 0.597 | 0.229 | 0.347 |
| Avg | **0.092** | **0.210** | 0.214 | 0.339 | 0.131 | 0.262 | 0.132 | 0.260 | 0.130 | 0.259 | 0.100 | 0.218 | 0.101 | 0.225 | 0.297 | 0.382 | 0.222 | 0.328 | 0.175 | 0.303 | 0.200 | 0.313 | 0.277 | 0.365 | 0.582 | 0.597 | 0.222 | 0.293 |
| **Weather** 12.5% | **0.025** | **0.045** | 0.057 | 0.141 | 0.047 | 0.101 | 0.039 | 0.084 | 0.041 | 0.107 | 0.027 | 0.051 | 0.026 | 0.047 | 0.140 | 0.220 | 0.037 | 0.093 | 0.037 | 0.072 | 0.031 | 0.076 | 0.296 | 0.379 | 0.176 | 0.287 | 0.036 | 0.095 |
| 25% | **0.029** | **0.052** | 0.065 | 0.155 | 0.052 | 0.111 | 0.048 | 0.103 | 0.064 | 0.163 | 0.029 | 0.056 | 0.030 | 0.054 | 0.147 | 0.229 | 0.042 | 0.100 | 0.038 | 0.074 | 0.035 | 0.082 | 0.327 | 0.409 | 0.187 | 0.293 | 0.042 | 0.104 |
| 37.5% | **0.031** | **0.057** | 0.081 | 0.180 | 0.058 | 0.121 | 0.057 | 0.117 | 0.107 | 0.229 | 0.033 | 0.062 | 0.032 | 0.060 | 0.156 | 0.240 | 0.049 | 0.111 | 0.039 | 0.078 | 0.040 | 0.091 | 0.406 | 0.463 | 0.172 | 0.281 | 0.047 | 0.112 |
| 50% | **0.034** | **0.062** | 0.102 | 0.207 | 0.065 | 0.133 | 0.066 | 0.134 | 0.183 | 0.312 | 0.037 | 0.068 | 0.037 | 0.067 | 0.164 | 0.249 | 0.053 | 0.114 | 0.042 | 0.082 | 0.046 | 0.099 | 0.431 | 0.483 | 0.195 | 0.303 | 0.054 | 0.123 |
| Avg | **0.030** | **0.054** | 0.076 | 0.171 | 0.055 | 0.117 | 0.052 | 0.110 | 0.099 | 0.203 | 0.032 | 0.059 | 0.031 | 0.057 | 0.152 | 0.235 | 0.045 | 0.104 | 0.039 | 0.076 | 0.038 | 0.087 | 0.365 | 0.434 | 0.183 | 0.291 | 0.045 | 0.108 |
| 1st Count | **48** | | 0 | | 0 | | 0 | | 0 | | 0 | | 0 | | 0 | | 0 | | 0 | | 0 | | 0 | | 0 | | 0 | |

Table 17: Full results for the classification task. ∗. in the Transformers indicates the name of ∗former. We report the classification accuracy (%) as the result. The standard deviation is within 0.1%.

| Datasets / Models | Classical methods | | | RNN | | TCN | | Transformers | | | | | | | | MLP | | TimesNet |
|---|---|---|---|---|---|---|---|---|---|---|---|---|---|---|---|---|---|---|
| | DTW (1994) | XGBoost (2016) | Rocket (2020) | LSTM (1997) | LSTNet (2018) | LSSL (2022) | TCN (2019) | Trans. (2017) | Re. (2020) | In. (2021) | Pyra. (2021a) | Auto. (2021) | Station. (2022a) | FED. (2022) | ETS. (2022) | Flow. (2022) | DLinear (2023) | LightTS (2022) | **(Ours)** |
| EthanolConcentration | 32.3 | 43.7 | 45.2 | 32.3 | 39.9 | 31.1 | 28.9 | 32.7 | 31.9 | 31.6 | 30.8 | 31.6 | 32.7 | 31.2 | 28.1 | 33.8 | 32.6 | 29.7 | 35.7 |
| FaceDetection | 52.9 | 63.3 | 64.7 | 57.7 | 65.7 | 66.7 | 52.8 | 67.3 | 68.6 | 67.0 | 65.7 | 68.4 | 68.0 | 66.0 | 66.3 | 67.6 | 68.0 | 67.5 | 68.6 |
| Handwriting | 28.6 | 15.8 | 58.8 | 15.2 | 25.8 | 24.6 | 53.3 | 32.0 | 27.4 | 32.8 | 29.4 | 36.7 | 31.6 | 28.0 | 32.5 | 33.8 | 27.0 | 26.1 | 32.1 |
| Heartbeat | 71.7 | 73.2 | 75.6 | 72.2 | 77.1 | 72.7 | 75.6 | 76.1 | 77.1 | 80.5 | 75.6 | 74.6 | 73.7 | 73.7 | 71.2 | 77.6 | 75.1 | 74.6 | 78.0 |
| JapaneseVowels | 94.9 | 86.5 | 96.2 | 79.7 | 98.1 | 98.4 | 98.9 | 98.7 | 97.8 | 98.9 | 98.4 | 96.2 | 99.2 | 98.4 | 95.9 | 98.9 | 96.2 | 96.2 | 98.4 |
| PEMS-SF | 71.1 | 98.3 | 75.1 | 39.9 | 86.7 | 86.1 | 68.8 | 82.1 | 82.7 | 81.5 | 83.2 | 82.7 | 87.3 | 80.9 | 86.0 | 83.8 | 75.1 | 88.4 | 89.6 |
| SelfRegulationSCP1 | 77.7 | 84.6 | 90.8 | 68.9 | 84.0 | 90.8 | 84.6 | 92.2 | 90.4 | 90.1 | 88.1 | 84.0 | 89.4 | 88.7 | 89.6 | 92.5 | 87.3 | 89.8 | 91.8 |
| SelfRegulationSCP2 | 53.9 | 48.9 | 53.3 | 46.6 | 52.8 | 52.2 | 55.6 | 53.9 | 56.7 | 53.3 | 53.3 | 50.6 | 57.2 | 54.4 | 55.0 | 56.1 | 50.5 | 51.1 | 57.2 |
| SpokenArabicDigits | 96.3 | 69.6 | 71.2 | 31.9 | 100.0 | 100.0 | 95.6 | 98.4 | 97.0 | 100.0 | 99.6 | 100.0 | 100.0 | 100.0 | 100.0 | 98.8 | 81.4 | 100.0 | 99.0 |
| UWaveGestureLibrary | 90.3 | 75.9 | 94.4 | 41.2 | 87.8 | 85.9 | 88.4 | 85.6 | 85.6 | 85.6 | 83.4 | 85.9 | 87.5 | 85.3 | 85.0 | 86.6 | 82.1 | 80.3 | 85.3 |
| Average Accuracy | 67.0 | 66.0 | 72.5 | 48.6 | 71.8 | 70.9 | 70.3 | 71.9 | 71.5 | 72.1 | 70.8 | 71.1 | 72.7 | 70.7 | 71.0 | 73.0 | 67.5 | 70.4 | **73.6** |

