# OpenReview forum: "TimesNet: Temporal 2D-Variation Modeling for General Time Series Analysis"
_ICLR.cc/2023/Conference — ICLR 2023 poster_

### Official Review · Reviewer_v92e · 2022-10-24

**Confidence:** 3
**Correctness:** 4
**Technical Novelty And Significance:** 3
**Empirical Novelty And Significance:** 3
**Recommendation:** 8

**Clarity, Quality, Novelty And Reproducibility:**

The proposed TimesBlock is novel. The paper is well organized, and well written.

**Details Of Ethics Concerns:**

A paper which has almost the same contents, organizations, was hound in Arxiv.
If the paper in Arxiv was sbmitted by the authors of this paper, I have come to know the authors, and I am concerned about whether this will be a problem as a double-brined rule.

**Strength And Weaknesses:**

Strength:
1. The proposed method solves the bottleneck of representation capability in 1D space and enables simultaneous representation of intra-period and inter-period variation in 2D space. While 1D time series could only represent variations between temporally adjacent time points, the proposed method offers the possibility of representing variations of the same phase in different periods. As the result, the proposed method allows the 2D kernel to simultaneously handle changes in adjacent time and changes between adjacent periods.

2. Good design for flexibility. The inception block can be replaced by any model of the type that uses a 2D kernel. In fact, the authors investigated its performance by replacing the inception block by other models such as ResNet and ResNeXt, etc.

3. The proposed method provides SOTA performance in five main tasks in time-series data analysis. While the models of previous studies differ in their strengths in each task, the proposed method almost consistently performs the best.


Concerns:
1. What does the dimension “C” of time series data correspond to in this study?

2. According to the appendix, the top-k was experimentally decided to be three (k=3). As far as I see the Fig.8, the performance was not so changed in the range between 1 and five. How carefully should the parameter k be decided? If a larger k is selected, is there any possibility to increase/decrease the performance?

3. Small concern. I found a paper, which has already submitted in Arxiv. The contents of both papers are almost same except for author area: one is anonymized, the other shows the author list. I hope this doesn't violate the double blind rule.

**Summary Of The Paper:**

The authors argue that 1D time series data are composed of multiple intraperiod-variations and interperiod-variations. Since there is a limit to capturing these variations as 1D time series representation, the authors transform the 1D time series into a set of 2D tensors based on multiple periods so that the analysis can be performed in a 2D space. The proposed method can adaptively discover multi-periodicity and extract complex temporal changes from 2D tensors. Experimental results show state-of-the-art performance in five mainstream time series analysis tasks, including short-term prediction, long-term prediction, imputation, classification, and anomaly detection.

**Summary Of The Review:**

The proposed method provides a new approach for 1D time series data by transform the 1D time series into a set of 2D tensors. It enables to use 2D kernels for analysis of both intraperiod- and interperiod-variations simultaneously. The inception block can be easily replaced by other existing models so that its flexibility is also high.

---

### Official Review · Reviewer_Re8p · 2022-10-24

**Confidence:** 3
**Correctness:** 3
**Technical Novelty And Significance:** 3
**Empirical Novelty And Significance:** 3
**Recommendation:** 6

**Clarity, Quality, Novelty And Reproducibility:**

In terms of clarity, the paper is well-organized and easy to follow. The proposed model uses FFT to identify periodicity and uses 2D vision backbones like inception to model the temporal dependency. Although both techniques are not novel, the idea of combining them to better utilize the multiple periodicities in time series to improve downstream task performance is reasonable and well executed in this paper. The paper discussed the implementation details and hyperparameter settings in appendices, so I personally believe the paper should be reproducible given that the source codes will be released.

**Strength And Weaknesses:**

Strengths:

- The idea of tackling the long-term dependency by utilizing the multiple periodicities in time series is reasonable and the use of FFT to extract the period from the frequency domain also makes sense.
- The empirical evaluation is comprehensive and promising results are shown.

Weaknesses:

- The periodicity in time series is an important assumption in this paper. Some time series data are not strictly periodic, but instead, they are composed of several repeating segments. In such cases, it is not clear if the proposed method still works.
- It is not clear if the proposed model can be applied to a dataset containing multiple datasets but with different period lengths for each data sample. For example, the time series data for patients in hospitals.


A minor detail: In the line before Eq. (1), "Fast Fourier Transformer (FFT)" should be "Fast Fourier Transform (FFT)".

**Summary Of The Paper:**

This paper focuses on modeling the temporal variations in the time series data. Specifically, based on the observation that there are often multiple periodicities in time series data, the paper proposes to divide a single multi-dimensional time series into several sets of smaller pieces of the time series, each with a different periodicity. Then, the inner-period variance and intra-period variance could be modeled together with a CNN backbone, like inception as used in this paper. Empirical evaluations are conducted with multiple datasets on forecasting, imputation, classification, and anomaly detection tasks.

**Summary Of The Review:**

Overall, I think this paper proposes a reasonable idea to tackle the long-term dependency issue in time series data modeling tasks and the solution is also sound. The empirical evaluations also show impressive results.

---

### Official Review · Reviewer_gBbZ · 2022-11-01

**Confidence:** 4
**Correctness:** 3
**Technical Novelty And Significance:** 2
**Empirical Novelty And Significance:** 3
**Recommendation:** 6

**Clarity, Quality, Novelty And Reproducibility:**

The presentation of this paper is relatively clear, though some clarification is needed as mentioned above.
Extensive experiments show the competitive performance of the proposed method.

The main concern is about the novelty and soundness of the proposed method. First, incorporating different component information (e.g., trend, seasonality, levels, etc.) from time series into neural network-based models is not a new idea. The proposed method is particularly focused on multi-periodicity and mostly employs well-known techniques. Second, compared with baselines intended for capturing different patterns in time series (e.g., periodicity, trend, levels, autocorrelation, etc.), the proposed method focusing on periodicities can mostly outperform. Consequently, it raises the concern of whether this outperformance is generalizable. In other words, the proposed method would have been positioned to a more specific scenario of periodic time series, instead of the "general time series" as it is now.

**Strength And Weaknesses:**

Strength:

The proposed method takes into account the multiple frequencies that could exist in multi-variate time series.
The experimental evaluation is extensive on several different time series tasks and shows competitive performance.

Weakness:

Incorporating different component information from time series into neural network-based models is not a new idea, for instance, one of the very recent works is [1]. Meanwhile, the following issues would affect the assessment of the paper.

(a) Some math notation would be misleading. e.g., $X_{1D}$ is two-dimensional to represent multivariate time series.

(b) Figure 2 is confusing. According to Eq. (1), the frequency analysis is applied to the multivariate time series, while Figure 2 gives the impression that it is to the univariate time series.

Based on Eq. (5), the Inception is applied to a matrix of dimension $ (f_i * p_i) * C$, while the illustration of 2D kernel seems to show that one dimension of the input to the Inception CNN is the number of frequencies.

(c) Given the frequencies selected based on the magnitude, the efficacy of adaptive aggregation Eq. (6) is unclear. It would be good to have some ablation study on this.

(d) During the training, the frequency analysis is applied to individual data instances or the batch of instances? Would the different granularities of this operation affect the performance?

(e) The truncate operation in Eq.(5) looks a bit unconvincing, given that it could blindly get rid of useful information depending on the dimension size relative to the dimension T.

(f)  In Table 11, the result of the baseline DLinear seems to be worse than that in the original paper [2]. It would be good to elaborate on it.

[1] Woo, Gerald, et al. "ETSformer: Exponential Smoothing Transformers for Time-series Forecasting." arXiv preprint arXiv:2202.01381 (2022).

[2] Zeng, Ailing, et al. "Are Transformers Effective for Time Series Forecasting?." arXiv preprint arXiv:2205.13504 (2022).



**Summary Of The Paper:**


This paper focuses on temporal variation modeling and proposes a representation method intended for incorporating multiple intraperiod- and interperiod-variations.
For each selected frequency, the proposed method generates a two-dimensional representation of the original time series and applies a convolutional network to the representation.
The proposed method is evaluated on several time series tasks including long/short-term forecasting, imputation, classification, and anomaly detection.

**Summary Of The Review:**


This paper focuses on temporal variation modeling and proposes a representation method intended for incorporating multiple intraperiod- and interperiod-variations. The proposed method is based on some existing ideas. It is evaluated on several time series tasks including long/short-term forecasting, imputation, classification, and anomaly detection, and shows competitive performance. The main concern is about the novelty and soundness of the proposed method, and probably a more appropriate positioning of the work.

---

> ### Comment · Reviewer_gBbZ · 2022-11-13
> **Authors' replies make the paper clear. The methodology-wise concern remains.**
>
> Thanks for the authors' reply and revision. It makes the paper clear.
>
> In my understanding, the essential idea of TimesNet can be summarized as follows: expanding each univariate time series of a multivariate one into a 2D matrix by segmenting 1d time series based on given periodicities, structuring periodicity-wise 3D tensors (2d matrixes plus the dimension of variables) to represent the multivariate time series,  and then applying a 2D kernel from computer vision to the 3D tensors.
>
> As for the argument that "our novelty in transforming the 1D series into 2D space, which can bridge the time series analysis with vision backbones.", considering there are massive works on applying neural networks used in computer vision to temporal data without the need of transforming the data, e.g.,  temporal convolution neural networks,  probabilistic Transformers, etc, the contribution of "bridge the time series analysis with vision backbones" is not substantial in my opinion.
>
> The multiperiodicity-based transformation essentially amounts to segmenting the time series into multiple subsequences and stacking them into a 2D matrix. Methodolothywise, this type of transformation seems practical and straightforward.

---

### Public Comment · ~chen_yuting1 · 2023-03-23
**Is it convenient for the authors to disclose the replies?**

Dear authors：

It's a very excellent work.
 And I am very interested in several questions raised by the reviewers during the review process and would like to see the authors' answers.
Is it convenient for you to disclose the replies?
Thanks!

---

> ### Author Response · Authors · 2023-03-23
> **Thanks for your interest**
>
> Hi, Thanks for your interest in TimesNet. You can send me an email for further discussion.

---

### Decision · Program_Chairs · 2023-01-20

**Decision:**

Accept: poster

**Justification For Why Not Higher Score:**

While the method is new, it is one of many that deal with time series, and the results aren't quite groundbreaking enough to warrant a spotlight.

**Justification For Why Not Lower Score:**

The contribution is valuable (new method + solid experiments), and likely to be of interest to the segment of the ICLR community that does research in the area of time series.

**Metareview: Summary, Strengths And Weaknesses:**

The paper presents a time series method that detect patterns of periodicity by transforming 1D time series into 2D tensors that are modeled by 2D kernels. The method is novel, and its ability to deal with multiple resolutions is likely to be useful in many practical applications beyond the benchmarks used. The transformation makes it possible to use deep learning architectures designed for vision tasks. This is an advantage, though not as large an advantage as it would have been prior to the invention of transformers, as  implied by reviewer gBbZ. The flexibility of the method is another nice feature. The experimental evaluation was considered convincing by all reviewers.
Overall, this is a good paper, showing a new method to handle trends of different frequencies in time series, with demonstrated practical applicability.


**Note From Pc:**

if the above contains the word "oral" or "spotlight" please see: "oral" presentation means -> notable-top-5% and "spotlight" means -> notable-top-25%. As stated in our emails, we are disassociating presentation type from AC recommendations

**Summary Of Ac-Reviewer Meeting:**

N/A